# An atypical basement membrane forms a midline barrier during left-right asymmetric gut development in the chicken embryo

Cora Demler[1†], John C Lawlor[1†], Ronit Yelin[2], Dhana Llivichuzcha-Loja[1], Lihi Shaulov[3], David Kim[1], Megan Stewart[1], Frank K Lee[1‡], Natalia Shylo[4], Paul A Trainor[4,5], Thomas M Schultheiss[2], Natasza A Kurpios[1*]

[1]Department of Molecular Medicine, College of Veterinary Medicine, Cornell University, Ithaca, United States; [2]Department of Genetics and Developmental Biology, Rappaport Faculty of Medicine, Technion – Israel Institute of Technology, Haifa, Israel; [3]Rappaport Faculty of Medicine, Technion – Israel Institute of Technology, Haifa, Israel; [4]Stowers Institute for Medical Research, Kansas City, United States; [5]Department of Anatomy and Cell Biology, University of Kansas Medical Center, Kansas City, United States

*For correspondence:
nk378@cornell.edu

†These authors contributed equally to this work

‡Deceased

Competing interest: The authors declare that no competing interests exist.

## eLife Assessment

This study reports the **fundamental** discovery of a novel structure in the developing gut that acts as a midline barrier between left and right asymmetries. Some of the evidence supporting the dynamics, composition, and function of this novel basement membrane in the chick is **solid**, some is even **convincing**, but investigation of its origin and impact on asymmetric organogenesis remains challenging and is not yet conclusive. This careful work is of broad relevance to patterning mechanisms, the importance of the extracellular matrix, and laterality disorders.

**Abstract** Correct intestinal morphogenesis depends on the early embryonic process of gut rotation, an evolutionarily conserved program in which a straight gut tube elongates and forms into its first loops. However, the gut tube requires guidance to loop in a reproducible manner. The dorsal mesentery (DM) connects the gut tube to the body and directs the lengthening gut into stereotypical loops via left-right (LR) asymmetric cellular and extracellular behavior. The LR asymmetry of the DM also governs blood and lymphatic vessel formation for the digestive tract, which is essential for prenatal organ development and postnatal vital functions including nutrient absorption. Although the genetic LR asymmetry of the DM has been extensively studied, a divider between the left and right DM has yet to be identified. Setting up LR asymmetry for the entire body requires a *Lefty1*+ midline barrier to separate the two sides of the embryo, without it, embryos have lethal or congenital LR patterning defects. Individual organs including the brain, heart, and gut also have LR asymmetry, and while the consequences of left and right signals mixing are severe or even lethal, organ-specific mechanisms for separating these signals remain poorly understood. Here, we uncover a midline structure composed of a transient double basement membrane, which separates the left and right halves of the embryonic chick DM during the establishment of intestinal and vascular asymmetries. Unlike other basement membranes of the DM, the midline is resistant to disruption by intercalation of Netrin4 (Ntn4). We propose that this atypical midline forms the boundary between left and right sides and functions as a barrier necessary to establish and protect organ asymmetry.

## Introduction

Humans and most other animals are bilaterians—animals whose left and right external features can be mirrored—but often the internal organs have striking left-right (LR) asymmetries. For example, in humans the heart resides on the left side of the thoracic cavity, the liver is predominantly on the right, and the spleen is on the left. Even paired organs like the lungs can show LR asymmetries—the left human lung has two lobes while the right lung has three. The left and right sides of the body are specified early in development, after the anterior/posterior and dorsal/ventral axes have been established (*McCain and McClay, 1994*; *Danos and Yost, 1995*). This patterning relies heavily on the expression of Sonic hedgehog (*Shh*) upstream of Nodal on the left side, and the repression of these genes by Activin on the right side (*Levin et al., 1995*). The localization of SHH-producing cells to the left is accomplished by nodal flow (mouse [*Nonaka et al., 1998*; *Nonaka et al., 2002*], zebrafish [*Essner et al., 2005*]. and *Xenopus* [*Schweickert et al., 2007*] embryos) or rotational cell movements around the node (chicken and pig embryos) (*Gros et al., 2009*), as well as cell death at the embryonic midline that may be a consequence of its abundant extracellular matrix (ECM) (*Maya-Ramos and Mikawa, 2020*).

Establishing the vertebrate LR body axis depends on a midline barrier to separate side-specific diffusible signals (*Bisgrove et al., 2000*; *Yamamoto et al., 2003*; *Meno et al., 1996*; *Meno et al., 1998*; *Yoshioka et al., 1998*; *Bisgrove et al., 1999*). This is achieved with a highly specific expression pattern of an inhibitor, Lefty1, at the center of the embryo which prevents the diffusible, left-sided signal NODAL from crossing to the embryo's right side (*Yamamoto et al., 2003*; *Meno et al., 1996*; *Meno et al., 1998*; *Yoshioka et al., 1998*; *Bisgrove et al., 1999*). Sixty percent of Lefty1-knockout mouse embryos die in utero and an additional 20% die before weaning, suffering from left isomerism of the lungs (in other words, both lungs have left lung lobation) and positional defects of the heart and the major vessels leading into/out of it (*Meno et al., 1998*). Other important examples of laterality defects are seen in conjoined twins, in which an embryo divides partially at the primitive streak stage (days 13–14 of gestation for humans) (*Kaufman, 2004*). In laterally conjoined (dicephalus) twins which form when the two primitive streaks are parallel, the left side of one embryo and the right side of the other are connected without the LEFTY1 barrier in between. Consequently, these conjoined twins often exhibit LR defects (*Levin et al., 1996*; *Tisler et al., 2017*). This underscores that separation of left and right signals is fundamental in early development for setting up correct placement and LR asymmetric patterning of individual organs including the heart and gut (*Essner et al., 2005*; *Bisgrove et al., 2000*; *Desgrange et al., 2018*; *Duboc et al., 2015*; *Desgrange et al., 2020*). Of all the organs with LR asymmetry, only the brain is known to harbor an organ-specific midline barrier (*Cavalcante et al., 2002*; *Kullander et al., 2001*; *Brose et al., 1999*; *Kidd et al., 1998*; *Erskine et al., 2000*; *Neugebauer and Yost, 2014*; *Katori et al., 2017*). Thus, midline barriers may have broad developmental significance for the embryo and its organs, yet very few such structures have been characterized.

Even organs that do not have obvious LR differences in the adult develop as a result of conserved LR asymmetric morphogenesis. The intestine is an excellent model for this, especially given the relatively simple tubular structure of the organ itself. During development, the intestine grows to great lengths (about 8 m in adult humans) (*Hounnou et al., 2002*), and this long tube must be looped to fit into the body cavity in a stereotypical, species-specific way (*Stevens and Hume, 1998*; *Savin et al., 2011*). When the developmental program directing gut looping is perturbed, as is the case for 1 in 500 infants who have congenital malrotation of the gut, there is an increased risk of volvulus, a lethal self-strangulation of the gut that requires immediate pediatric surgical intervention (*Torres and Ziegler, 1993*).

The gut tube does not loop autonomously. Rather, gut looping is directed by the neighboring dorsal mesentery (DM) (*Figure 1A*), a thin mesodermal organ that connects the gut to the rest of the body and through which intestinal blood and lymphatic vessels traverse (*Savin et al., 2011*; *Mahadevan et al., 2014*; *Hu et al., 2021*; *Hecksher-Sørensen et al., 2004*). The left and right sides of the DM take on different properties at the molecular, cellular, and extracellular levels which are critical to initiate asymmetric gut looping and vascular morphogenesis (*Mahadevan et al., 2014*; *Kurpios et al., 2008*; *Davis et al., 2008*; *Welsh et al., 2013*; *Sivakumar et al., 2018*; *Sanketi et al., 2022*). Gut tilting is the symmetry-breaking event that initiates asymmetric gut looping, which occurs at embryonic day 10.5 (E10.5) in the mouse and Hamburger-Hamilton stage 19–21 (HH19–21) (*Hamburger and Hamilton,*

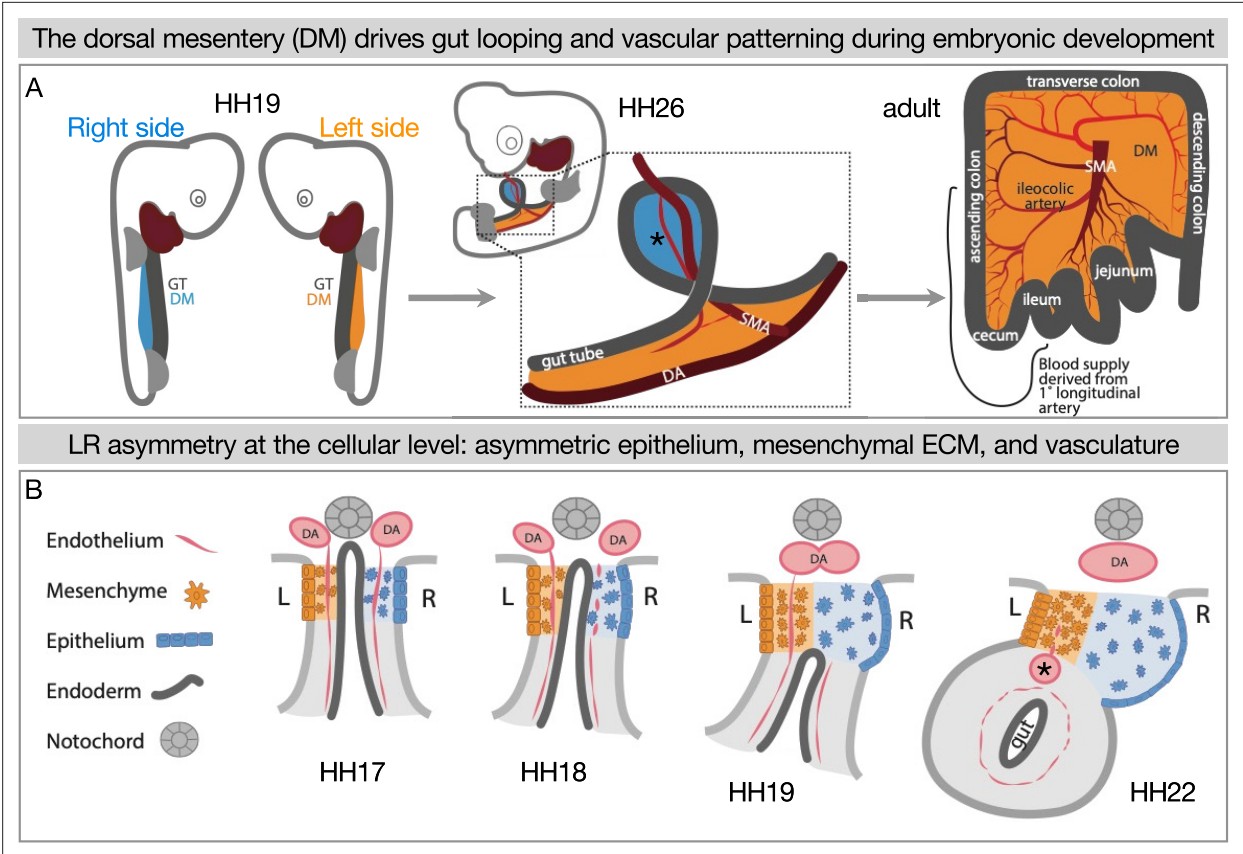

**Figure 1.** Left-right (LR) asymmetry in the DM is critical for proper gut looping and vascular patterning. (**A**) Asymmetries in the DM drive the formation of the first and subsequent gut loops. Concurrently, the vasculature is being patterned in the DM. The 1° longitudinal artery (*) gives rise to the ileocolic artery, which provides a significant portion of the adult intestine with critical blood flow. (**B**) Hamburger-Hamilton stage 17 (HH17): The DM has cellular symmetry. HH18 and 19: The right mesenchyme begins expanding and the right epithelial cells elongate. The right-sided endothelial cells (vascular precursors) begin to disperse and leave the compartment, while the left-sided endothelium is maintained to become the future gut arteries. HH22: The asymmetric forces have pushed the gut tube to the left. A left-sided blood vessel, the 1° longitudinal artery (*) has formed. GT = gut tube, DM = dorsal mesentery, DA = dorsal aorta, SMA = superior mesenteric artery.

1951) in the chicken embryo (*Figure 1*; *Davis et al., 2008*). Gut tilting is driven by the condensation of the ECM in the left DM and an expansion of the ECM on the right (*Figure 1B*; *Kurpios et al., 2008*). Blood vasculature also develops asymmetrically. Although endothelial precursors exist on both the left and right DM prior to tilting, as the asymmetries are established the right-sided endothelial cells emigrate rather than forming vessels (*Mahadevan et al., 2014*). Only a left-sided gut artery develops further (*Figure 1B*), which goes on to supply blood to a significant portion of the adult intestine (*Figure 1A*; *Mahadevan et al., 2014*). Interestingly, these right-sided endothelial cells emigrate dorsally and ventrally, but they cannot cross directly over to the left side (*Mahadevan et al., 2014*). This might indicate the presence of a barrier against cell migration at the midline.

The differences in the left and right sides of the DM are well understood at multiple levels of biology, but the maintenance of these asymmetries has not been explored. The classes of genes with asymmetric expression in the DM include transcription factors, ECM components, and genes which encode diffusible signals that could spread within the DM compartments to create morphogen gradients across the LR axis. Instead of an LR gradient, however, there is a sharp delineation between left and right DM in gene expression and protein localization, as well as LR asymmetric distribution of extracellular glycosaminoglycans, such as hyaluronan (*Mahadevan et al., 2014*; *Kurpios et al., 2008*; *Davis et al., 2008*; *Welsh et al., 2013*; *Sivakumar et al., 2018*; *Sanketi et al., 2022*). Not surprisingly, perturbing even just one gene's asymmetric expression patterns in the DM leads to aberrant gut looping patterns and abnormal vascular lesions (*Mahadevan et al., 2014*; *Welsh et al., 2013*; *Sivakumar et al., 2018*; *Sanketi et al., 2022*). Thus, we hypothesized that a barrier exists in the center

of the DM to segregate cells and diffusible signals to the left and right compartments. Here, we show that an atypical basement membrane forms a boundary between left and right cell populations and limits diffusion across the DM midline.

## Results

### Cells of left- and right-origin meet but do not mix in the DM

The DM mesenchyme forms by bilateral epithelial-to-mesenchymal transition (EMT) and ingression of coelomic epithelium, which is derived from the splanchnic mesoderm; the left DM comes from the left coelom and the right DM arises from the right coelom (*Figure 2A and B*; *Hecksher-Sørensen et al., 2004*; *Kurpios et al., 2008*; *Davis et al., 2008*). This was visualized by injecting early chicken embryos with DiI and DiO into the right and left coelomic cavities, respectively (*Figure 2C–F*), or by electroporating each side with plasmids encoding different fluorophores (*Figure 2G and H*). While the left and right cells meet at the middle of the DM, they never cross over or mix (*Mahadevan et al., 2014*; *Kurpios et al., 2008*; *Davis et al., 2008*; *Welsh et al., 2013*; *Sanketi et al., 2022*; *Welsh et al., 2015*; *Arraf et al., 2016*; *Arraf et al., 2020*). This striking separation occurs despite the lack of a visible histological boundary between the two sides as shown by H&E staining at HH20 and HH21, when the left is condensing and the right is expanding to drive the leftward gut tilting (*Figure 2K and L*). Early in development the endoderm effectively separates the left and right splanchnic meso-derm (*Figure 2I and J*), but once the DM forms and the endoderm descends it is likely important to continue separating the two sides until asymmetries can be established (*Figure 1B*). In support of this, we have previously shown that when cell-cell adhesion is interrupted in the left DM, the cells become more dispersed (*Kurpios et al., 2008*; *Welsh et al., 2013*) and extend filopodia over toward the right side, suggesting pathogenic cell migration (*Welsh et al., 2013*). Thus, the critical separation between left and right cells in the DM can be disrupted, necessitating a mechanism for protecting these asymmetries.

### The DM midline is not marked by Lefty1, but by laminin

The early embryo uses a molecular barrier of *Lefty1*-expressing cells to separate laterality signals so the LR axis is established correctly (*Yamamoto et al., 2003*; *Meno et al., 1996*; *Meno et al., 1998*; *Yoshioka et al., 1998*; *Bisgrove et al., 1999*). To see if this mechanism is adapted by the intestine for establishment of its laterality, we performed *Lefty1* RNA in situ hybridization on both early (HH9) and later (HH19) stages. While *Lefty1* was expressed at the midline of early embryos as expected (*Figure 3A*), it was not expressed at the midline of the DM at HH19 (*Figure 3B*). This indicates that a different mechanism must be at work during the establishment of gut asymmetries. Interestingly, scanning electron microscopy (SEM) data showed a fibrous matrix between the noto-chord and endoderm where the DM will later develop, suggesting that ECM may separate the two sides before they coalesce into the DM (*Figure 3C and D*). Consistent with this hypothesis, basement membranes are found in other biological contexts where a barrier is needed, such as in the skin or around blood vessels (*Yurchenco, 2011*). Basement membranes are dense ECM requiring laminin, collagen IV, nidogen, and perlecan and/or agrin (both heparan sulfate proteoglycans) with a large variety of other components that can be integrated to create specific 'flavors' of basement membrane tuned to different barrier contexts (*Yurchenco, 2011*). We therefore postulated that the DM midline has a physical barrier consisting of basement membrane, rather than a *Lefty1* molecular barrier.

To test this hypothesis, we visualized the basement membrane marker laminin by immunofluo-rescence (IF) for laminin alpha 1 (Lama1) at developmental stages where DM asymmetries are being established. As expected, this marked several typical, single-layer basement membranes underlying polarized cells, such as around the notochord (*Figure 3E and F*; *Bancroft and Bellairs, 1976*; *Saraga Babić, 1990*), coelomic epithelium (*Figure 3E–H*; *Magro and Grasso, 1995*), and gut endoderm (*Figure 3E–H*; *Simon-Assmann et al., 1998*). We also observed scattered laminin staining in the DM mesenchyme, which is a consequence of those cells carrying basement membrane fragments with them after EMT and ingression from the coelomic epithelium (*Horejs, 2016*; *Hu et al., 2013*). Inter-estingly, we identified a previously uncharacterized atypical double basement membrane within the DM. At HH18, when cellular DM asymmetries are first being initiated at the level of the midgut (which forms the small intestine), laminin IF marked an oval-shaped structure just ventral to the notochord

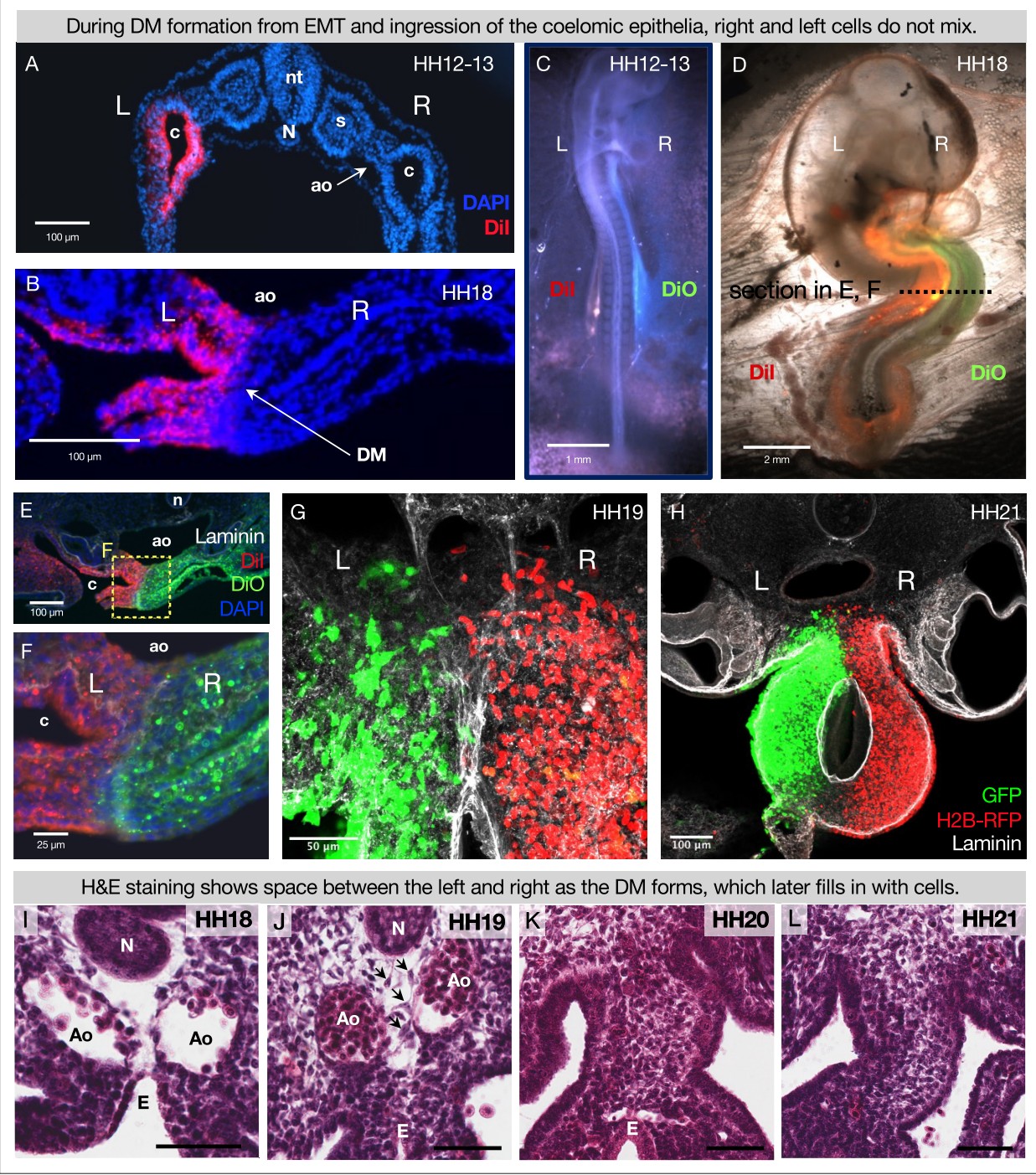

**Figure 2.** During DM formation from epithelial-to-mesenchymal transition (EMT) and ingression of the coelomic epithelia, right and left cells do not mix. When the coelomic cavity is injected with DiI at Hamburger-Hamilton stage 12–13 (HH12–13), n=5 (**A**), the labeled cells give rise to the mesenchymal and epithelial cells of the DM on the corresponding side of the embryo, n=5 (**B**). When DiI and DiO are injected at HH12–13 into left and right coeloms, respectively, n=6 (**C**), labeled cells are still segregated at HH18, n=6 (**D, E, F**). The same results are found when cells are labeled by electroporation with pCAG-GFP (left) and pCl-H2B-RFP (right) (**G, H**), both when the midline is continuous (HH19 n=3), (**G**) and once it has disappeared (HH21 n=3, **H**). (**I–L**) H&E staining of the DM at HH18 n=5 (**I**) shows 'empty space' between the notochord, endoderm, and dorsal aortae. At HH19 n=5 (**J**), this space gains some cells (arrows), and the space is completely filled in by HH20 n=4 (**K**) and HH21 n=3 (**L**). Scale bars = 60 µm. nt = neural tube, c = coelom, ao = aorta, N = notochord, s = somite, DM = dorsal mesentery, L = left, R = right.

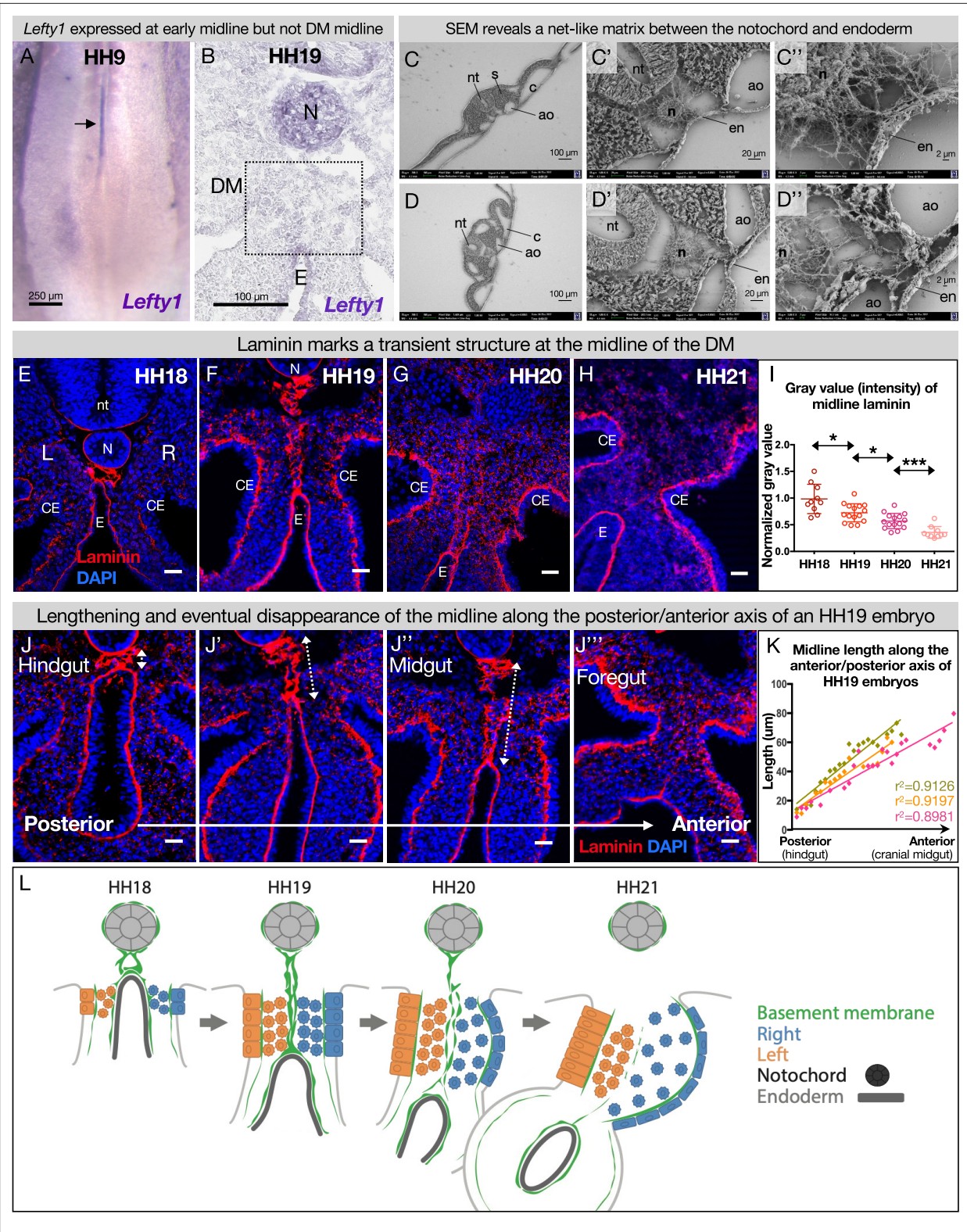

**Figure 3.** The dorsal mesentery (DM) midline is marked by laminin immunofluorescence. (**A**) *Lefty1* expression is seen at the embryonic midline of this Hamburger-Hamilton stage 9 (HH9) embryo n=4 (arrow). (**B**) *Lefty1* expression is not seen at the midline of the DM (dashed box) of an HH19 embryo (n=4) (notochord = positive control). (**C, D**) Scanning electron microscopy (SEM) images of a fixed embryo at HH15–16 show fibrous extracellular matrix (ECM) between the notochord and endoderm. (**C**) is from a more posterior axial level than (**D**). (**E–H**) Time course of midline dynamics from HH18–21, marked by laminin. Scale bars = 25 μm. (**I**) Quantification of the intensity of laminin immunofluorescence over development, normalized to laminin

*Figure 3 continued on next page*

*Figure 3 continued*

staining around the neural tube. Biological replicates: HH18 n=2, HH19 n=3, HH20 n=3, and HH21 n=2. Each dot represents one image quantified. Statistical analyses are unpaired Welch's t-tests. HH18–19: p=0.0188, HH19–20: p=0.0118, HH20–21: p=0.0003. Additionally, there is a significant (p<0.0001) linear trend among the means from HH18 to HH21 with a slope of –0.1019, $r^2$=0.5906. (**J**) Midline appearance from hindgut to foregut in an HH19 embryo, marked by laminin. Scale bars = 25 µm. (**K**) Quantification of DM midline length (dashed line) of three HH19 embryos, from the separation of the notochord and endoderm (hindgut) to the fusion of the aortae (foregut, coinciding with midline fragmentation). (**L**) Model of DM midline time kinetics. N = notochord, E/en = endoderm, Ao = aorta, nt = neural tube, c=coelomic cavity, CE = coelomic epithelium, L = left, R = right.

The online version of this article includes the following source data and figure supplement(s) for figure 3:

**Source data 1.** Summary table of embryo stages, statistical testing, and graph data for laminin intensity and midline length in *Figure 3*.

**Figure supplement 1.** Scanning electron microscopy (SEM) of cross-sections of chicken embryos at different stages of midline development.

**Figure supplement 2.** The required basement membrane components nidogen and perlecan co-localize with laminin at the midline.

**Figure supplement 3.** Pseudotime kinetics of the midline along the anterior-posterior axis.

**Figure supplement 4.** The midline is conserved in a squamate and follows a similar trajectory of degradation.

and dorsal to the gut endoderm (*Figure 3E*). No cells were seen within this structure as shown by a lack of nuclear staining (*Figure 3E*) and empty space in SEM (*Figure 3C and D*, *Figure 3—figure supplement 1*). As the DM elongates and asymmetries become more apparent (HH19), this midline structure lengthened, forming two parallel lines connecting the endoderm and notochord (*Figure 3F*). One stage later (HH20), the midline was still present but began to appear fragmented (*Figure 3G*). By HH21, the asymmetries of the DM are established—the right DM has expanded, the left DM has condensed, the gut has tilted to the left, and vascular precursor cells have been driven out of the right side (*Figure 1B*; *Mahadevan et al., 2014*; *Sanketi et al., 2022*). Surprisingly, the midline disappeared by this stage, while laminin IF underlying the coelomic epithelium and endoderm remained intense (*Figure 3H*). The lengthening of the midline and its subsequent loss occurred in an anterior-to-posterior wave down the embryonic gut tube (*Figure 3J and K*). Consequently, anterior sections of younger embryos (i.e. HH12–13) had similar midline structures to posterior, older sections (i.e. HH18–19).

## The DM midline consists of a transient, true basement membrane

The combination of laminin, nidogen, perlecan (or agrin), and collagen is the foundation of all basement membranes (*Yurchenco, 2011*). To further characterize the nature of the ECM at the DM midline, we did IF staining for nidogen and perlecan, confirming co-localization with laminin at the midline barrier (*Figure 3A*, *Figure 3—figure supplement 2*). This further illustrates that the DM midline consists of basement membrane. This basement membrane structure is conserved in the squamate veiled chameleon, *C. calyptratus*, which exhibits a similarly transient double basement membrane at the midline of the DM from approximately the 7-somite stage to the 29-somite stage (*Figure 3—figure supplement 4*; *Diaz et al., 2019*).

In addition to the four foundational basement membrane components, a myriad of other proteins, proteoglycans, and glycoproteins can assemble onto the basement membrane (*Jayadev and Sherwood, 2017*). Consequently, there is a vast variety of 'flavors' of basement membrane with different physical properties and different signals to adjacent cells about polarity, migration, or other behaviors (*Jayadev and Sherwood, 2017*). A common basement membrane constituent is fibronectin, which is best known for its role in the provisional matrix during wound healing (*Clark et al., 1982*). At our developmental stages of interest, fibronectin localized to the dorsal aorta which often coincides with the most ventral part of the midline (*Figure 3—figure supplement 3*). Together, we model the midline as a transient double basement membrane that bisects the DM during developmental stages when critical asymmetries are being established (*Figure 3L*).

## The midline does not originate from the left or right DM

Although we have established the time kinetics of DM midline formation, the origin of this structure remains elusive. The midline is sandwiched between mesenchymal cells from the left and right DM, unbiased to either the left or the right side (*Figure 2F and G*). Intriguingly, mesenchymal cells like these are not usually competent to construct an organized basement membrane (*Yurchenco, 2011*; *Glentis et al., 2014*). Mesenchymal cells can secrete matrix components (*Simon-Assmann et al.,*

*1998*), but the organization of these components into a basement membrane is dependent on the presence of cell surface anchors which are characteristic of tissues like polarized epithelium or endothelium, not mesenchymal cells (*Glentis et al., 2014*). In the case of LAMA1, it is known that this protein is secreted by the epithelia in the developing intestine, not the mesenchyme (*Simon-Assmann et al., 1998*). Indeed, RNA in situ hybridization for *Lama1* did not show enriched expression in the mesenchyme at the DM midline (*Figure 4A–C*). Moreover, if the cells adjacent to the midline were secreting and organizing the basement membrane, we would expect these cells to be polarized like the cells of the coelomic epithelium or endoderm. As expected, the left coelomic epithelium was polarized at HH19 relative to its basement membrane as quantified by Golgi staining with GM130 (*Figure 4D and E*, *Figure 4—figure supplement 1*; *Welsh et al., 2013*). However, GM130 staining and quantification showed that cells immediately to the left or right of the midline have random orientation (*Figure 4D and E*, *Figure 4—figure supplement 1*). Together, these data allow us to rule out a mesenchymal origin for the DM midline.

## During endodermal descent, endodermal cells are not left behind to form the midline

Given that the DM midline connects the notochord and endoderm, both of which have their own basement membranes and are very closely associated early in development, we hypothesized that one or both of these structures contribute to midline production (*Figure 3C, E, and J*, *Figure 3—figure supplement 1*). As the embryo grows and the DM elongates, the distance between the notochord and endoderm increases (*Figures 2I–J and 3E–H*), and the midline is found between them as a double line of basement membrane. Thus, we hypothesized that as the endoderm descends ventrally, it undergoes EMT and leaves behind basement membrane-carrying cells to form the midline. To test this, we developed a method to specifically target the endoderm using DNA electroporation (*Figure 4F*). Briefly, we injected pCAG-GFP plasmid underneath HH14–15 embryos and applied an electric pulse such that the endodermal cells would take up the DNA (*Figure 4F*), so we could lineage trace endodermal cells during DM formation. Interestingly, embryos isolated at HH19 and HH21 showed that GFP-labeled cells remain restricted to the endoderm—there were no GFP+ mesenchymal cells present in the DM (*Figure 4G and H*). This indicates that the DM midline is not formed from EMT of basement membrane-carrying endodermal cells.

## The notochord is not sufficient for DM midline formation

To test whether the notochord is sufficient for midline formation, we performed notochord transplant experiments. In brief, the notochord was removed from an HH12–15 embryo. An incision was made in a stage-matched recipient embryo adjacent to the neural tube and the donor notochord was inserted into this slit (*Figure 4I*). Embryos continued to develop until isolation at HH19. These transplants were done to the left and right sides of different embryos (*Figure 4J/K and L/M* respectively). RNA in situ hybridization was performed for *chordin* to ensure that the transplanted notochord was alive and functioning (*Figure 4J and L*; *Sasai et al., 1994*; *Streit et al., 1998*). Laminin IF did not reveal a secondary midline-like structure associated with the ectopic notochord (*Figure 4K and M*) while the normal midline was unaffected (*Figure 4K and M*, white arrows). This result is seen regardless of whether the transplants are done to the embryo's left or right side. From this, we conclude that the notochord is not sufficient for the formation of DM midline.

## The DM midline is resistant to degradation by Netrin4

Laminin matrices are susceptible to competitive disruption by the matrix protein Netrin4 (NTN4) (*Schneiders et al., 2007*; *Reuten et al., 2016*). NTN4 has very high binding affinity for laminin gamma subunits, such that NTN4 can prevent the formation of new laminin networks, which are the foundation upon which other basement membrane components assemble and can also disrupt existing laminin networks (*Schneiders et al., 2007*; *Reuten et al., 2016*). *Ntn4* is not endogenously expressed in the DM (data not shown), which allows us to use it as a tool to target basement membranes in the DM. As expected, when we overexpressed *Ntn4* on either side of the DM by electroporation, we perturbed the basement membrane underlying the coelomic epithelium and depleted the scattered laminin staining in the mesenchyme that results from EMT creating the DM (*Figure 5B, D, and E* vs. controls *Figure 5A and C*; *Horejs, 2016*; *Hu et al., 2013*). Intriguingly, the DM midline basement membrane

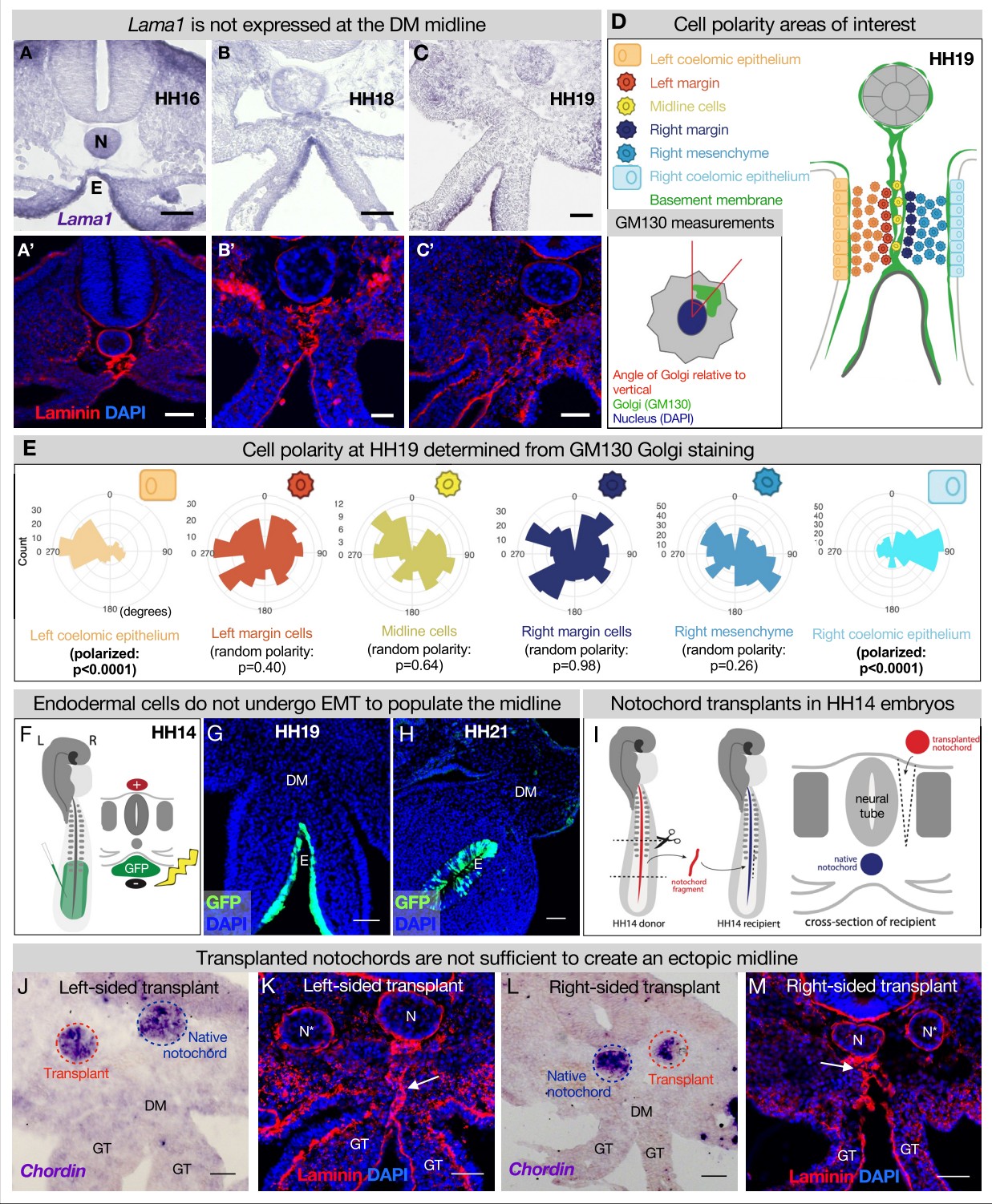

**Figure 4.** The midline basement membrane is not made by the DM mesenchyme or epithelial-to-mesenchymal transition (EMT) of the endoderm, and the notochord is not sufficient for midline formation. (**A–C**) *Lama1* RNA in situ hybridization and adjacent sections with laminin immunofluorescence (IF) staining at Hamburger-Hamilton stage 16 (HH16) n=3, HH18 n=9, and HH19 n=10 (**A'–C'**). Scale bars = 50 μm. (**D, E**) Cell polarity analysis from GM130 staining shows that the mesenchymal cells immediately to the left or right of the midline ('left/right margin') and within the double membrane ('midline cells') have random polarity, as do the cells of the right mesenchyme (random polarization control), in contrast to the strong apical-basal polarity in cells of the left coelomic epithelium. Five embryos were used for these quantifications. Number of cells per circle histogram: left coelomic epithelium = 209, left margin = 346, midline cells = 118, right margin = 413, right mesenchyme = 514, right coelomic epithelium = 295. (**F**) Electroporation mix

*Figure 4 continued on next page*

*Figure 4 continued*

containing pCAGEN-GFP plasmid was injected under an HH14/15 embryo and then electroporated to specifically target the endoderm. Lineage tracing endoderm-derived cells to HH19 n=8 (**G**) and HH21 n=2 (**H**) by pCAGEN-GFP electroporation of the endoderm. (**I**) Model of notochord transplant method. A piece of notochord (red) was isolated from an HH14 donor embryo. In a stage-matched recipient, a cut was made adjacent to the neural tube and the donor notochord was inserted into this slit. (**J, L**) RNA in situ hybridization for Chordin to mark the native notochord (blue dashed circle) and transplanted notochord (red dashed circle). (**K, M**) Laminin immunohistochemistry to mark basement membrane including the midline (white arrow). Notochords are marked with an N (native notochord) and N* (transplanted notochord). (**J, K**) are from the same embryo, as are (**L, M**). n=8, scale bars = 50 μm. GT = gut tube, DM = dorsal mesentery, E=endoderm.

The online version of this article includes the following source data and figure supplement(s) for figure 4:

**Source data 1.** Summary table of embryo stages, statistical testing, and graph data for cell polarity in *Figure 4*.

**Figure supplement 1.** GM130 staining for polarity analysis.

remained visibly intact in all embryos with lateral (*Figure 5B and D*) and bilateral (*Figure 5E*) *Ntn4* overexpression in the DM. Similarly, endodermal *Ntn4* overexpression caused much less disruption of the endodermal or midline basement membranes (*Figure 5G* vs. control *Figure 5F*) when compared to its effect on coelomic epithelium basement membrane. This was true even when the electroporations were done much earlier in development (HH10 and HH12–13, data not shown). This suggests that the midline and endoderm may have basement membranes of the same 'flavor', possibly pointing to a common origin. In contrast, the basement membrane beneath the coelomic epithelium may be more susceptible to NTN4 disruption because of the prior EMT-induced breaks in the basement membrane (*Horejs, 2016*; *Hu et al., 2013*) or a different protein composition.

## DM midline is a barrier against diffusion

Genes including Cxcl12 (*Mahadevan et al., 2014*; *Sivakumar et al., 2018*) and Bmp4 (*Sanketi et al., 2022*) which encode diffusible signals are expressed asymmetrically in the DM (*Figure 6A and B*). So too are genes encoding enzymes that are secreted into the ECM, like the HA-modifying enzyme TSG6 (*Sivakumar et al., 2018*). The expression domains of these genes have a sharp boundary at the midline, since left and right cells do not mix. However, the secreted protein products of these genes may be able to diffuse across the DM if their movement is not limited (*Figure 6C*). We know that experimentally mixing left and right signals is detrimental to gut tilting and vascular patterning (*Mahadevan et al., 2014*; *Kurpios et al., 2008*; *Davis et al., 2008*; *Welsh et al., 2013*; *Sivakumar et al., 2018*; *Sanketi et al., 2022*)—e.g., ectopic expression of pro-angiogenic Cxcl12 on the right side results in an aberrant vessel forming on the right (*Mahadevan et al., 2014*). Moreover, when the CXCR4 receptor antagonist AMD3100 (MW = 502.78) is introduced to the left DM, it abolishes vascular development on the left. However, when the same drug is introduced to the right DM, the left-sided vascular development remains intact (*Mahadevan et al., 2014*). This phenomenon suggests a barrier against diffusion.

To test if the basement membrane structure at the midline is forming a functional barrier against diffusion, we injected 3 kDa fluorescent dextran directly into the right side of the DM (*Figure 6D–G*). When these injections are performed at stages where the midline is intact (HH19), movement of dextran through the tissue was limited to the right side (n=4/4 embryos) (*Figure 6D*). When the basement membrane midline appears fragmented (HH20), these injections produced mixed results—in some embryos (n=2/9), diffusion across the midline was prevented and in others (n=7/9) the dextran was able to move into the left mesenchyme of the DM (*Figure 6E*). Finally, at stages where no organized basement membrane structure remains at the midline (HH23), diffusion of dextran was always permitted across the entire width of the DM (n=7/7) (*Figure 6F*).

To confirm our finding that the basement membrane structure at the midline forms a barrier against diffusion, we utilized a BODIPY-tagged version of AMD3100 (*Poty et al., 2015*), delivered via soaked resin beads surgically inserted into the left coelomic cavity (*Figure 6H–L*). The ratio of average AMD3100-BODIPY intensity in the right DM vs. the left DM was below 0.5 when the midline is intact (HH19, n=4/4), indicating little diffusion across the DM (*Figure 6J*). When no midline remains at a later stage (HH21, n=4/4), this ratio significantly rises to near one, indicating diffusion of the drug is not impeded when the midline basement membrane structure is absent (*Figure 6J*). Collectively, these data suggest that the basement membrane structure at the midline forms a transient functional barrier against diffusion (*Figure 6M*).

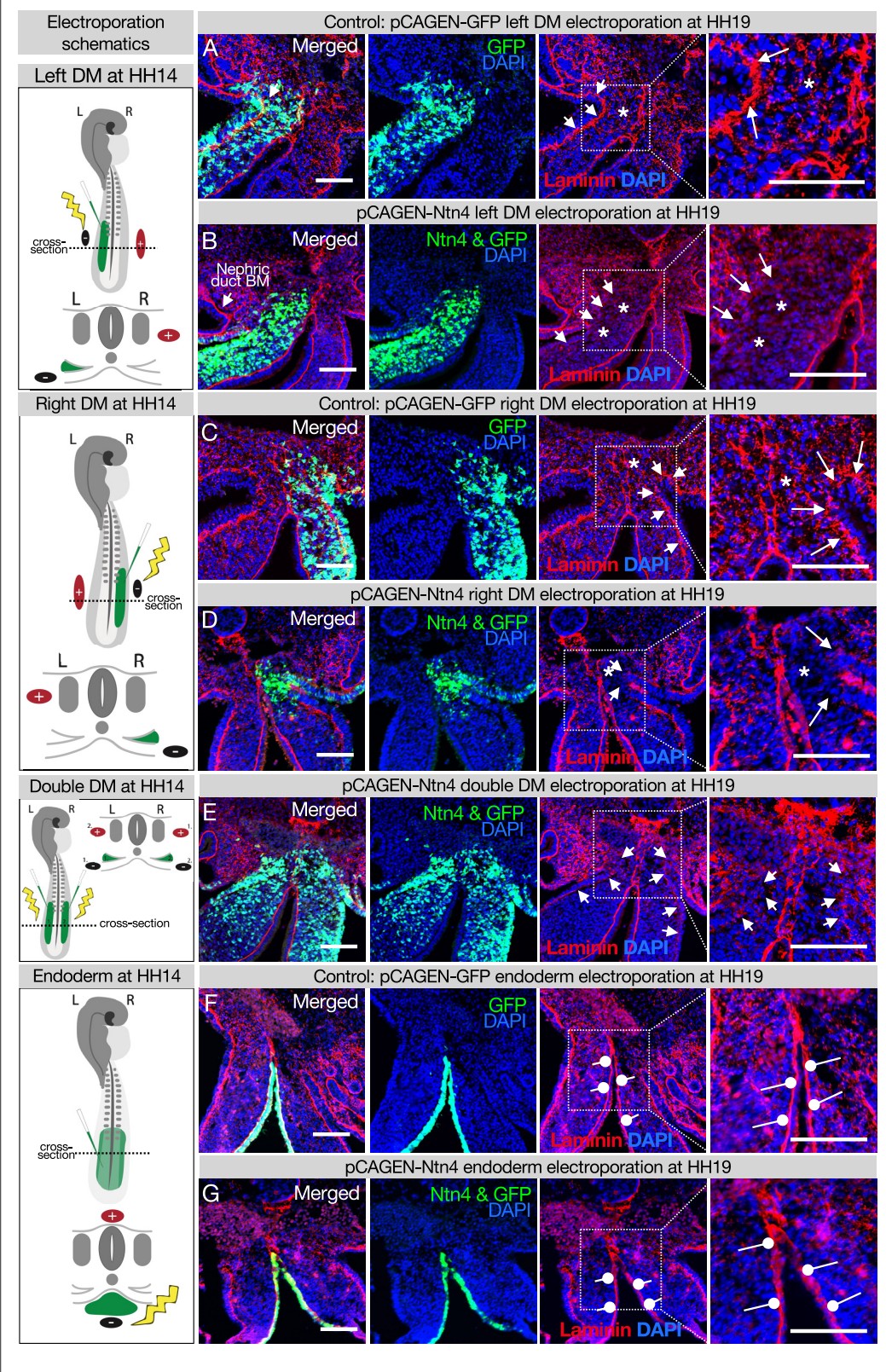

**Figure 5.** Ectopic expression of Netrinn4 by electroporation visibly affects the basement membrane underlying the coelomic epithelium, but not that underlying the endoderm, or the midline. (**A, C**) Electroporation of the left (**A**) or right (**C**) dorsal mesentery (DM) with the control, pCAG-GFP, had no effect on the basement membrane of the coelomic epithelium (arrows). Left n=6. Right n=4. (**B, D**) Electroporation of mouse Netrin4 (pCAGEN-Ntn4) and

*Figure 5 continued on next page*

*Figure 5 continued*

pCAG-GFP into the left (**B**) or right (**D**) DM disrupted the basement membrane underlying the coelomic epithelium (arrows) and scattered basement membrane in the mesenchyme (asterisks). Left n=5. Right n=5. The intact laminin staining in **B** is the basement membrane (BM) of the nephric duct. (**E**) Double DM electroporations also disrupt the coelomic epithelium (arrows) but the midline appears intact despite being contacted by Ntn4+ cells (n=3). (**F, G**) Electroporation of the endoderm directly with pCAG-GFP or pCAGEN-Ntn4 does not visibly affect the basement membrane underlying the endoderm (open round pointers). Control n=5. Ntn4 n=8. The midline appears unaffected by any of these perturbations. Scale bars = 100 μm.

## Discussion

Establishing the left and right body plan in the early embryo is a fundamental part of development and this process depends on the presence of a Lefty1+ midline barrier. Individual organs, too, have LR asymmetries, but the brain has the only known organ-specific midline barrier, where commissural axons of the brain and spinal cord are tightly controlled by midline-localized guidance and repulsion cues including FGFs, SLIT/ROBO signaling, EFNB3, heparan sulfate proteoglycans, and the Rac-specific GTPase-activating protein α-chimaerin (*Cavalcante et al., 2002*; *Kullander et al., 2001*; *Brose et al., 1999*; *Kidd et al., 1998*; *Erskine et al., 2000*; *Neugebauer and Yost, 2014*; *Katori et al., 2017*). The developing intestine has a similar need for separation between left and right cells and signals, but it seems to accomplish this by a different mechanism—an atypical basement membrane at the midline.

This basement membrane may separate cells that have ingressed from the right and left coelomic epithelia (*Carmona et al., 2013*). since these cells do not mix at the midline (*Figure 2*). It may also prevent mixing of diffusible signals. While the diffusion of a given signal depends on the tissue context (*Müller et al., 2013*). some morphogens (like Nodal) can induce effects at distances of 200 μm or greater (*Müller and Schier, 2011*). The HH19 DM is only about 150 μm across, not a prohibitive distance for diffusible signals like BMPs, TGFβ (*Sanketi et al., 2022*), and CXCL12 (*Mahadevan et al., 2014*) to cross between the left and right DM. Thus, we hypothesized that the DM may need a barrier at the midline to segregate these signals. In support of this, we showed that the midline limits diffusion of dextran from right to left, which suggests that it also blocks the movement of endogenous diffusible signals. Moreover, with a molecular weight of just 3 kDa, dextran's inability to cross the midline indicates that diffusible proteins of typical weight (such as CXCL12 at 10 kDa and BMP4 at 34 kDa) are also unable to cross. We previously showed evidence that even the 502.78 Da drug AMD3100 was prevented from moving to the other side of the DM (*Mahadevan et al., 2014*). In line with these findings, we have now employed a BODIPY-labeled form of AMD3100 (*Poty et al., 2015*). Similar to dextran, our results show BODIPY diffusion across the midline when the basement membrane is intact. However, when no basement membrane remains at the midline, BODIPY diffusion was permitted across the entire DM width. While the midline could not be degraded by *Ntn4* overexpression, future studies may reveal tools for the selective destruction of this basement membrane to better understand its function in gut laterality.

This work adds a new facet to our knowledge of basement membrane form and function. Basement membranes play many critical barrier functions in the embryo and adult, usually found as a single layer that underlies polarized epithelial or endothelial cell layers such as those lining the intestines, encircling blood vessels, or enveloping muscle cells, adipocytes, or Schwann cells (*Yurchenco, 2011*). As such, null mutations in genes encoding basement membrane components often result in embryonic lethality and postnatal pathologies (*Bader et al., 2005*; *Miner et al., 2004*; *Smyth et al., 1999*; *Gatseva et al., 2019*; *Pozzi et al., 2017*; *Yao, 2017*). However, a role for basement membrane in establishing LR asymmetry has not been described previously.

The basement membrane we describe here is atypical in its double membrane structure, which raises interesting questions about its formation. We showed that the midline is not produced by the mesenchymal cells of the DM (*Figure 4*), and that the notochord is not sufficient for its synthesis (*Figure 4*). Instead, we consider the endoderm. Upon electroporation of the endoderm with GFP, GFP-positive cells were not detected in the DM later in development but remained restricted to the endoderm (*Figure 4F–H*), indicating that the midline does not form from EMT of basement membrane-carrying endodermal cells. We can also eliminate endodermal death as a possible mechanism, because there was no appreciable cell death observed by TUNEL staining at the midline during the stages of interest (data not shown). Nor does the midline form in the same way as other double basement membranes,

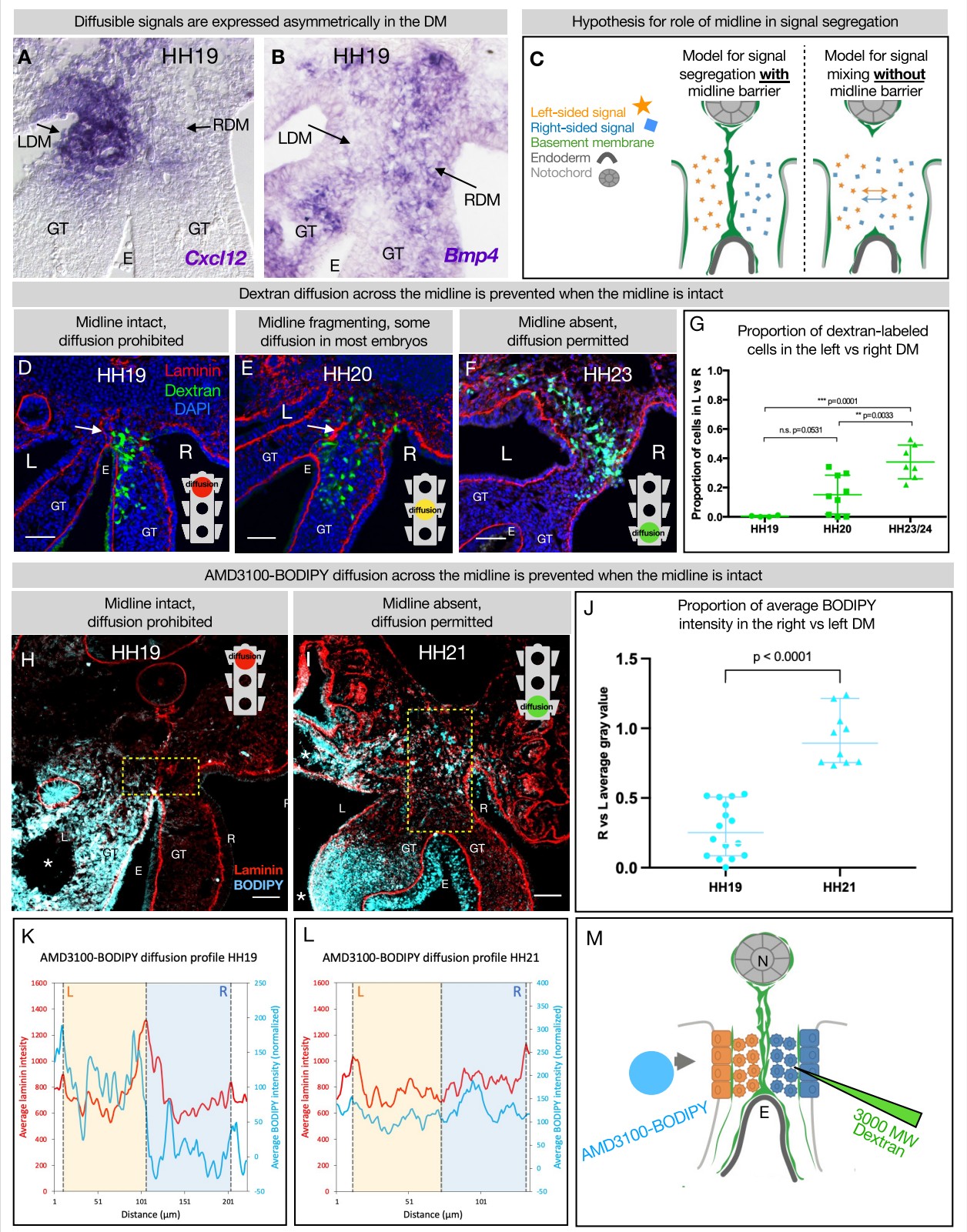

**Figure 6.** The dorsal mesentery (DM) midline serves as a barrier against diffusion. (**A, B**) Genes encoding diffusible signals including *Cxcl12* and *Bmp4* are expressed asymmetrically in the DM at Hamburger-Hamilton stage 19 (HH19). (**C**) Hypothesis for the role of the midline in limiting diffusion of left and right signals across DM. (**D**) At HH19, the midline is intact (white arrow) and diffusion of 3000 MW dextran (green) is limited to the right side (n=4/4). (**E**) At HH20, the midline (white arrow) has begun to fragment. Diffusion across the midline is prohibited in some embryos (n=2/9) but permitted in

*Figure 6 continued on next page*

*Figure 6 continued*

others (n=7/9). (**F**) At later stages when the midline has disappeared, diffusion is allowed through the DM (n=7/7). (**G**) Proportion of dextran-labeled cells in the left vs. right DM, with unpaired t-test. (**H**) At HH19, the midline is intact and diffusion of AMD3100-BODIPY is limited to the left side (n=4). Dashed yellow box indicates quantified area. (**I**) At HH21 when the midline has disappeared, diffusion is allowed through the DM (n=3). Dashed yellow box indicates quantified area. (**J**) Proportion of BODIPY intensity in the right vs. left DM, with unpaired t-test. Each dot represents one image quantified. (**K, L**) Profile plot of average BODIPY intensity across the DM within the dashed yellow boxes in H and I, with left and right compartments of the DM overlayed. (**M**) Schematic of dextran injections into the right DM and AMD3100-BODIPY beading into the left DM. Scale bars = 50 µm. LDM = left dorsal mesentery. RDM = right dorsal mesentery. GT = gut tube. E = endoderm. L=left. R=right. N=notochord. DA = dorsal aorta.

The online version of this article includes the following source data for figure 6:

**Source data 1.** Summary table of embryo stages, statistical testing, and graph data for dextran and BODIPY diffusion in *Figure 6*.

which arise from the meeting of the basal sides of two tissues (*Keeley and Sherwood, 2019*; *Pastor-Pareja, 2020*). This is classically illustrated in the kidney glomerulus between epithelial podocytes and endothelial cells (*Pastor-Pareja, 2020*; *Miner, 2012*; *Naylor et al., 2021*). and in the blood-brain barrier between endothelial cells, pericytes, and astrocytes (*Keeley and Sherwood, 2019*; *Daneman and Prat, 2015*). Instead in the case of the DM midline barrier, the apical sides of the endodermal cells are facing each other.

We hypothesize that the midline forms when basement membrane is left behind as the endoderm descends ventrally during normal development, as if it were a 'scar' of where the endoderm was previously (*Figure 7*). During early development the notochord is embedded within the endoderm and a basement membrane covers the two structures. Only later are the two structures separated by a full basement membrane, pointing to a strong connection between the two (*Fausett et al., 2014*; *Jurand, 1974*). The medial migration of the aortae and coelomic epithelia could provide tissue forces that push the notochord and endoderm apart, leaving behind basement membrane at the DM midline. The observation that the basement membranes of both the endoderm and midline were resistant to disruption by NTN4 could also support the idea that the endoderm is responsible for making both basement membranes (*Figure 7*). In support of this, we know that the basement membrane underlying the gut endoderm does not co-migrate with the intestinal epithelial cells as they move from the proliferative intestinal crypts to the tip of the villus over the course of 3–6 days in the adult; the basement membrane is instead left in place. (*Trier et al., 1990*). One way to test this hypothesis would be to 'trace' endodermal basement membrane by electroporation of a tagged basement membrane

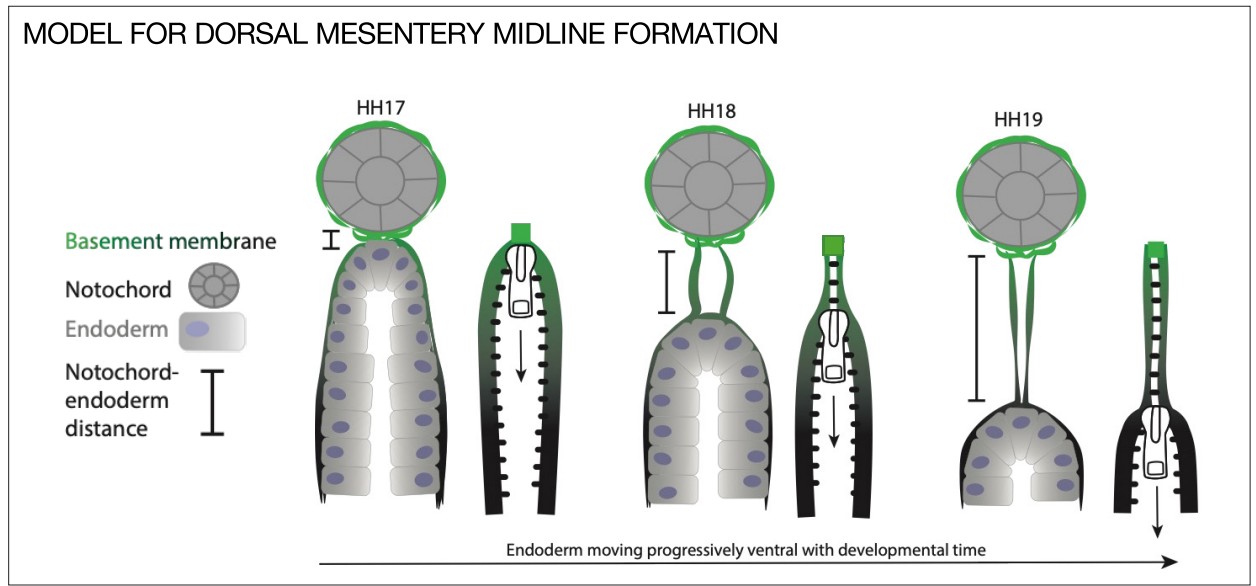

**Figure 7.** Model of endoderm descending hypothesis for midline formation. We hypothesize that as the endoderm moves ventrally and the distance between the notochord and endoderm grows, basement membrane from the endoderm may be left behind. This can be compared to a zipper where each side is the basement membrane underlying the endoderm, and when the zipper pull (tip of endoderm) moves downward, the basement membrane behind it pulls closer together.

component, as has been done recently in *Caenorhabditis elegans* (**Walser et al., 2017**; **Naegeli et al., 2017**; **Matsuo et al., 2019**; **Jayadev et al., 2019**; **Keeley et al., 2020**; **Jayadev et al., 2022**) and *Drosophila* (**Morin et al., 2001**; **Ramos-Lewis et al., 2018**). However, these advances are only just now reaching the mouse system (**Tomer et al., 2022**; **Morgner et al., 2023**) and have not yet made their way into the chick embryo.

The endodermal basement membrane and midline's resistance to NTN4 disruption is particularly notable, as similar resistance has not been reported before. This suggests the presence of an unknown factor or modification that shields or stabilizes the laminin network from NTN4 binding. Investigating the mechanisms that stabilize basement membranes against such disruptions could be important for future developmental studies and research on basement membranes in the context of disease and cancer.

Although Lefty1 expression is not observed at the midline when this basement membrane is robust, it remains unclear whether midline Lefty1 expression is crucial for the construction or proper function of this structure. Future research could involve siRNA inhibition of Lefty1 expression at an early stage, followed by an evaluation of the midline structure's integrity at HH19. However, similar to perturbing global Nodal expression, it may be challenging to differentiate between the direct impact of Lefty1 loss on future midline cells, and the indirect impact on midline cells resulting from free Nodal diffusion throughout the early lateral mesoderm that would happen as a consequence of Lefty deficiency. Further investigation into the relationship between these two aspects of the midline barrier is important.

The midline barrier is set apart from many other basement membranes by its rapid disappearance. We consider that the loss of the DM midline barrier may be caused in part by 'stretching' of the basement membrane as the notochord and endoderm become increasingly separated due to the elongation of the DM (**Figure 3J**). Since the DM itself does not contribute basement membrane to the midline (**Figure 4C**) and the midline length increases quickly (**Figure 3J**), it is plausible that the midline may be pulled until it reaches its ultimate tensile strength (between 0.5 and 3.8 MPa in other examples of naturally occurring basement membranes) (**Jain et al., 2022**), and then breaks. This would suggest that midline breakdown is a passive consequence of embryo growth.

However, we also consider that there could be an active breakdown mechanism for the midline. The turnover of stable basement membranes occurs on the scale of weeks (**Trier et al., 1990**; **Decaris et al., 2014**), but the midline barrier appears to degrade over just 12–24 hr. In other contexts, basement membrane destruction can occur over a large area, such as by secretion of matrix metalloproteinases in metastasizing cancer (**Miyoshi et al., 2004**; **Miyoshi et al., 2005**), or more localized, as exhibited by invadopodia on immune or cancer cells before metastasis (**Sekiguchi and Yamada, 2018**; **Santiago-Medina et al., 2015**). Such localized basement membrane breakdown is seen in critical developmental processes, including mouth development in deuterostome embryos (**Dickinson and Sive, 2006**). The oral membrane includes a basement membrane that closes off the digestive system from the outside world. This basement membrane specifically disintegrates to rupture that membrane to open the early mouth cavity (**Dickinson and Sive, 2006**). In addition, a basement membrane divides the two halves of the embryonic brain, and must be broken down at the site of the corpus callosum for neurons to cross for inter-hemisphere communication in the cerebrum (**Hakanen and Salminen, 2015**; **Gobius et al., 2016**). Localized basement membrane dissolution is also critical for optic cup fusion, where a contact-dependent dissolution of basement membrane occurs between the two sides of the optic fissure (**Torres et al., 1996**; **Barbieri et al., 2002**). Coloboma, a congenital eye defect where some tissue is missing inside the eye resulting in an enlarged, irregular pupil, occurs when optic fusion arrests (**Patel and Sowden, 2019**; **ALSomiry et al., 2019**). It is worth noting that in all of these examples and the case of the midline, the location of basement membrane breakdown is very specific—neighboring basement membranes appear unaffected (**Figure 3H**). Together, these findings support the idea that basement membrane breakdown in the DM may not simply be a passive process but may be required by the embryo for later events in gut development or vascular patterning. We have not yet identified any matrix protease specific to the midline of the DM. Given that a mechanism for basement membrane breakdown is not well characterized in any of the above contexts, there is high potential for future investigation.

The DM midline may have effects on development beyond the intestine. It has long been known that signals produced from axial structures in the embryo are critical for the establishment of LR

asymmetry, but it has been difficult for researchers to pin down which 'midline structure' or 'midline signal' is actually responsible. For example, the heart (*Lohr et al., 1997*; *Chen et al., 1997*), lungs (*Arraf et al., 2016*), and kidneys (*James and Schultheiss, 2003*) all rely on 'dorsal midline structures' to develop properly. Aberrations from the loss of midline structures like the notochord include 'horse-shoe kidney', in which the kidneys stay close to the midline and fuse at their posterior end (*Natsis et al., 2014*). This is due to the lack of notochordal Shh signaling, but the 'midline barrier' that exists downstream from this Shh signaling remains a mystery (*Tripathi et al., 2010*).

The DM midline barrier may also play a role in vascular patterning, including aortic fusion. The aorta begins as two parallel tubes with an avascular zone in between (where the midline is) (*Garriock et al., 2010*), but progressively they fuse into one with an anterior-to-posterior wave until the level of the vitelline arteries (*Figure 3—figure supplement 3*; *Garriock et al., 2010*). Fusion of the aortae coincides with the fragmentation and disappearance of the midline, suggesting that these two processes may be inextricably linked. Proper timing of this fusion depends on carefully balanced levels of VEGF (*Jadon et al., 2023*). SHH (*Vokes et al., 2004*), and the anterior-to-posterior wave of downregulation of the BMP-inhibitory genes Chordin and Noggin from the notochord (*Garriock et al., 2010*; *Reese et al., 2004*; *Sato, 2013*). The precise mechanism of dorsal aorta fusion remains unknown, although there is evidence that VEGF signaling pulls VE-cadherin away from its cell-cell junctions (*Jadon et al., 2023*). This relocalization may be important for the remodeling of the aortic endothelium during fusion and may be linked to the breaking of the DM midline barrier.

Midline structures are critical for laterality to develop correctly. The notochord is certainly involved, particularly as the source of modulators of BMP and Hedgehog signaling. However, this does not seem to be the whole story, given that in each of these contexts the actual 'midline barrier' downstream from notochord signals has not been identified. It is possible that the DM midline basement membrane is key here, either for separating left and right signals or perhaps for binding signals (*Pozzi et al., 2017*) from the notochord to act as a buffer between each side. The midline barrier may also play a role in the rheology of the DM. Microindentation analyses at HH21 show that the condensed left DM is significantly stiffer than the expanded right DM, and that proper gut tilting is dependent on this difference being tightly regulated (*Sanketi et al., 2022*). It is possible that the midline barrier helps to segregate the stiffness-influencing components of each side (i.e. covalently modified HA on the right [*Sivakumar et al., 2018*], N-cadherin on the left [*Kurpios et al., 2008*]) and provides a 'wall' for the right side to push upon in order to swing the gut tube toward the left. It's important to note that left and right mesenchymal cells do not cross the midline to the opposite side of the DM. Likewise, during the formation of gut arteries, a small subset of gut vascular endothelial cells migrate from right to left, closely adjacent to the dorsal tip of the endoderm, but they do not traverse the midline (*Mahadevan et al., 2014*). Moreover, when the midline basement membrane is present and intact, N-cadherin expression is symmetric across the DM, but becomes asymmetric and left-specific upon midline disintegration (*Kurpios et al., 2008*). This may represent a shift in cell segregation mechanisms in the DM—while the midline is intact, the double basement membrane structure appears sufficient to segregate left and right cells. When the midline disappears, a new cellular separation mechanism is required to maintain asymmetric compartments and N-cadherin functions in this role.

Looping patterns of the midgut are stereotypical between individuals but vary between species (*Savin et al., 2011*). While differential growth rates between the gut tube and the DM have been identified as the primary means by which distinct gut looping patterns may be achieved (*Savin et al., 2011*), it is also possible that mesentery-specific changes such as modifications to the kinetics of asymmetry in the DM may impact looping patterns with potentially adaptive impact on diet and therefore niche utilization. The presence, permeability, and timing of degradation of the midline basement membrane may provide evolution with an additional means of fine-tuning looping patterns between species. We have found this basement membrane to be transiently present in the DM of veiled chameleon embryos from just after the 7-somite stage to the 29-somite stage (*Figure 3—figure supplement 4*; *Diaz et al., 2019*), suggesting this structure is conserved in reptiles and birds at a minimum, with intriguing heterochrony in degradation between species. In mammals, prior studies have observed midline laminin deposition in the mouse gut located between the separating notochord and endoderm, further suggesting conservation of a midline basement membrane in amniotes (*Li et al., 2007*; *Hajduk et al., 2012*).

Collectively, we have identified a novel midline barrier in the gut mesentery that is composed of an atypical double basement membrane that forms a boundary between the left and right sides and limits movement of diffusible signals and cells, at a stage when Lefty1 is no longer expressed at the midline. The DM midline also presents an opportunity to interrogate the fundamental mechanisms of basement membrane formation and degradation during vertebrate embryonic development, with implications for research on cancer metastasis. We posit that this midline is a distinct strategy for the critical separation of left and right signals and cells, key for establishing and maintaining LR asymmetry for healthy gut development.

# Materials and methods

## Key resources table

| Reagent type (species) or resource | Designation | Source or reference | Identifiers | Additional information |
|---|---|---|---|---|
| Antibody | Anti-laminin alpha 1 rabbit polyclonal (1° ab) | Sigma | L9393 | 1:100 |
| Antibody | Anti-laminin 1 mouse monoclonal (1° ab) | DSHB | 3H11 | 1:10 |
| Antibody | Anti-perlecan mouse monoclonal (1° ab) | DSHB | 5C9 | 1:10 |
| Antibody | Anti-nidogen mouse monoclonal (1° ab) | DSHB | 1G12 | 1:10 |
| Antibody | Anti-fibronectin mouse monoclonal (1° ab) | DSHB | VA1(3) | 1:5 |
| Antibody | Anti-fibronectin mouse monoclonal (1° ab) | DSHB | B3/D6 | 1:30 |
| Antibody | Anti-fibronectin rabbit polyclonal (1° ab) | Sigma | F3648 | 1:400 |
| Antibody | Anti-GM130 mouse monoclonal (1° ab) | BD Biosciences | 610822 | 1:250 |
| Antibody | Alexa Fluor 568 goat anti-rabbit (2° ab) | Invitrogen | A-11031 | 1:500 |
| Antibody | Alexa Fluor 647 donkey anti-rabbit (2° ab) | Invitrogen | A32795 | 1:500 |
| Antibody | Alexa Fluor 488 goat anti-mouse (2° ab) | Invitrogen | A32723 | 1:500 |
| Other | DAPI | Thermo Fisher | D1306 | 1:2000, nuclear DNA counterstain |
| Chemical compound, drug | Dextran, Fluorescein, 3000 MW, lysine fixable, anionic | Thermo Fisher | D3306 | |
| Chemical compound, drug | AMD3100-Bodipy | *Poty et al., 2015* | | 5 mg/ml |
| Chemical compound, drug | CM-DiI | Invitrogen | C7000 | |
| Chemical compound, drug | SP-DiO | Invitrogen | D7778 | |
| Recombinant DNA reagent | Plasmid for chordin riboprobe (chicken) | Cliff Tabin lab | T691 | |
| Recombinant DNA reagent | Plasmid for lefty1 riboprobe (chicken) | Cepko/Tabin lab | T607 | |
| Recombinant DNA reagent | pCAGEN (plasmid) | Connie Cepko | RRID:Addgene_11160 | |
| Recombinant DNA reagent | pCAG-GFP (plasmid) | Connie Cepko | RRID:Addgene_11150 | |
| Recombinant DNA reagent | pCI-H2B-RFP (plasmid) | Addgene | RRID:Addgene_92398 | |

*Continued on next page*

*Continued*

| Reagent type (species) or resource | Designation | Source or reference | Identifiers | Additional information |
|---|---|---|---|---|
| Recombinant DNA reagent | Ntn4-AP-His (plasmid) | Addgene | RRID:Addgene_71980 | |
| Sequence-based reagent | F primer for chicken LAMA1 riboprobe from cDNA | This paper | | ACGGAGAGTTTGGCAGATGA |
| Sequence-based reagent | R primer for chicken LAMA1 riboprobe from cDNA | This paper | | ATCCTGAGCCCAAATCCCAA |
| Sequence-based reagent | 5′ primer for cloning Ntn4 coding region out of RRID:Addgene_71980 and into pCAGEN (XhoI and NotI) | This paper | | ATGCCTCGAGATATCgccaccatggggagctg |
| Sequence-based reagent | 3′ primer for cloning Ntn4 coding region out of RRID:Addgene_71980 and into pCAGEN (XhoI and NotI) | This paper | | CTAGCGGCCGCGGATCCATCGATTATTA CACGCAGTCTCTTTTTAAGATGTGCA |
| Commercial assay or kit | PCR cloning kit (with pDrive plasmid) | QIAGEN | 231124 | |
| Other | Fertilized chicken eggs | Westwind Farms (Interlaken, NY, USA, http://chickenhawkfood.com). | | Eggs used for embryo manipulation and collection as described in Materials and methods |
| Other | Veiled chameleon eggs | Reptiles and Aquatics Facility at Stowers Institute for Medical Research | | Eggs used for embryo manipulation and collection as described in Materials and methods |
| Other | AG beads | Bio-Rad | 143-1255 | Resin beads for surgical implantation and drug diffusion as described in Materials and methods |

## Chicken embryo development and processing

Fertile chicken eggs were purchased from Westwind Farms (Interlaken, NY, USA, http://chicken-hawkfood.com). After 36–48 hr of incubation at 37°C, eggs were windowed by removing 8 ml of thin albumen with an 18 ½ gauge needle/10 ml syringe and cutting an oval in the side of the shell, then covering the opening with clear packing tape and returning the egg to the incubator. Once at the desired stage, embryos were isolated in cold 1× PBS and fixed overnight in 2% paraformaldehyde (PFA) at 4°C, followed by PBS washes. Embryos were prepared for cryo-embedding by putting them through graded sucrose solutions ending in 30% sucrose overnight at 4°C. Embryos were cryo-embedded in OCT (VWR 25608-930), sectioned to 15 μm, dried overnight, then stored at –80°C.

## Immunofluorescence

Cryosections were rehydrated in PBS then PBST (0.03% Tween-20), then blocked in 3% heat-inactivated goat serum (HIGS, Gibco 16210072) in PBST for 45 min at room temperature. Primary antibodies were diluted in blocking solution (3% HIGS in PBST) and incubated either for 45 min at room temperature or overnight at 4°C. After three PBST washes for 5 min each, secondary antibodies were incubated for 45 min at room temperature with 1:2000 dilution of DAPI added. PBST and PBS washes were done before mounting the slides with Prolong Gold anti-fade (Invitrogen P36930). Antigen retrieval pretreatment was necessary for GM130 (BD Biosciences 610822) immunofluorescent staining. Cryosections were rehydrated in water, then microwaved in 1:100 antigen retrieval solution (Vector Laboratories, H-3300) until nearly boiling. After incubating at 37°C wrapped in aluminum foil for 15 min and cooling for 10 min, slides were taken through the standard IHC protocol.

## RNA in situ hybridization

Section and wholemount RNA in situ hybridization was done using a modified protocol from Moisés Mallo as previously described (*Aires et al., 2019*).

## Dextran injections

Dextran injections into the DM were done using 3000 MW dextran conjugated to fluorescein (Thermo Fisher D3306) at a concentration of 10 mg/ml in 1× PBS with Fast Green dye added to better visualize

the solution during injection. This mixture was loaded into fine pulled glass capillary needles. A micro-injector with a foot pedal was set to 5 psi for 200 ms. Chicken embryos at the desired stage were prepared by removing the vitelline membrane. Injections were done to the right DM only, because embryos lie on their left sides from HH18 onward and only the right DM is accessible for injection. With the anterior/posterior axis of the embryo perpendicular to the needle and with the needle at a 25° angle, the body wall was gently pulled back so the needle could access the right side of the DM. The needle was gently pressed into the tissue until the embryo moved slightly from the force. Then, the foot pedal was pressed once to inject. Embryos were allowed to continue incubating for about 2 hr, then embryos were collected and fixed in 2% PFA overnight at 4°C. To screen for embryos with quality injections, embryos were cryo-embedded and sectioned. Any embryos with visible damage to the DM in these sections were excluded from further analysis.

## AMD3100-BODIPY beading

AMD3100-BODIPY was synthesized as previously described (*Poty et al., 2015*), and AG beads (Bio-Rad 143-1255) were soaked in 5 mg/ml AMD3100-BODIPY overnight nutating at 4°C. Drug-laden beads were inserted into the left coelomic cavity via a small incision, as previously described (*Mahadevan et al., 2014*).

## CM-DiI and SP-DiO injections

Five µl of a stock solution of CM-DiI or SP-DiO (1 µg/µl in EtOH) was diluted into 45 µl of prewarmed 0.3 M sucrose in single distilled water maintained at 37°C. Dye solutions were injected into the coelomic epithelium as previously described (*Arraf et al., 2016*).

## Electroporation

DM electroporations were performed as described previously (*Sivakumar et al., 2018*; *Sanketi and Kurpios, 2022*). Endodermal electroporations were performed with a similar method, but with the electroporation mix (plasmid of interest and/or pCAG-GFP, 1× PBS, 1× Fast Green, 1 mM MgCl$_2$, and 0.17% carboxymethylcellulose) injected into the empty space beneath the ventral side of the embryo while the negative electrode was held in place there. The positive electrode was placed directly above the negative electrode, centered along the neural tube, before the pulse was applied. pCAGEN-Ntn4 expression plasmid was constructed by cloning the full-length coding sequence out of mouse Ntn4-AP-His plasmid (RRID:Addgene_71980) using the primers in Key resources table and cloning into pCAGEN (RRID:Addgene_11160) with XhoI (NEB R0146) and NotI (NEB R0189). A second Ntn4 construct was also used, using a mouse Ntn4 (generously provided by Raphael Reuten) and cloned into the pMES vector (*Swartz et al., 2001*) with similar results.

## Notochord transplants

Notochord transplants were performed on HH12–15 embryos, using a method adapted from papers describing notochordectomies and notochord transplants (*Teillet and Le Douarin, 1983*; *Klessinger and Christ, 1996*; *Artinger and Bronner-Fraser, 1993*; *van Straaten et al., 1985*; *Straaten et al., 1988*; *Yamada et al., 1991*; *Pettway et al., 1990*). Note that notochordectomies, while potentially informative, were not done because damage to the endoderm is highly likely in those experiments. Since the endoderm is also potentially implicated in midline formation, an experiment that did not perturb that tissue was preferred. To prepare the donor notochord, the embryo was cut crosswise at the level of the vitelline arteries and close to the end of the tail. Clean 2–4 mm sections of notochord were used for the transplants. A sharp glass needle was used to make an incision in the recipient embryo along the anterior/posterior body axis between the neural tube and somites, deep enough so the ectopic notochord could sit next to the native notochord without puncturing the dorsal aorta. The donor notochord was pressed into place using a pair of dull glass needles or forceps. Embryos continued developing at 37°C until stage HH19.

## Scanning electron microscopy

Embryos used for *Figure 3C and D* were fixed in 4% PFA and 2% glutaraldehyde in PBS overnight at 4°C. Samples were then equilibrated in 5% sucrose/PBS for 1 hr at room temperature, 20% sucrose/PBS for 1 hr at room temperature, and finally 15% sucrose/7.5% gelatin/PBS at 37°C overnight.

Embryos were then embedded in plastic molds and frozen in liquid isopentane in a dry ice-ethanol bath. Cryosections of 10 μm in thickness were collected on poly-L-lysine-treated coverslips and incubated twice with fresh drops of PBS for 10 min at 37°C. Coverslips were then washed with 0.1 M cacodylate buffer and post-fixed with 0.1% osmium tetroxide. Following washes with deuterium-depleted water, the sections were dehydrated in graded ethanol series, critical point dried (Quorum K850), and sputter coated with 6 nm of chromium (Quorum Q150T). Samples were then viewed on Zeiss Ultra Plus HR Scanning Electron Microscope using the SE2 detector.

Embryos for SEM in *Figure 3—figure supplement 1* were fixed in 2% glutaraldehyde in 0.50 M cacodylate buffer (pH 7.4) at 4°C for 2 hr, then rinsed three times for 10 min each in 0.05 M cacodylate buffer. At this point, embryos were cut down to size, using sharp spring scissors to cut the embryo crosswise at the level of the midgut. Then, embryos were post-fixed in 1% osmium tetroxide in 0.05 M cacodylate buffer at 4°C for 1 hr, rinsed again in 0.05 M cacodylate buffer (3×10 min), dehydrated in an ethanol series of 25%, 50%, 70%, 95%, and 100% for 10 min each, and left in 100% ethanol overnight. The following day embryos were critical point dried in $CO_2$, soaking for 24 hr. Samples were mounted and silver paint was used for conductivity. These embryos were sputter-coated with gold palladium and imaged on a LEO 1550 (Keck SEM).

## Imaging, image processing, and quantifications

Brightfield and fluorescent images of tissue sections were taken on a Zeiss Observer Z1 with Apotome, an LSM880 Confocal multiphoton inverted microscope—i880 (Zeiss), or an LSM710 Confocal (Zeiss), or a ScanScope CS2. Stereoscopic images were taken on a SteREO Discovery.V12 (Zeiss). Images were processed using Fiji. Statistical analyses were done using GraphPad Prism.

Quantification of fluorescent intensity of laminin staining was done for five sum intensity projections per biological replicate. For each image, the width of the midline was averaged from three measurements. When no midline is discernible at HH21, the midline width for each image was replaced with the overall average HH20 midline width. Then, five profile plots of gray values (intensity) were obtained from orthogonal lines drawn across the midline. The gray values within the average membrane width centered around the local maximum gray value were averaged to produce a raw mean midline gray value for each image. This process was repeated for neural floor plate basement membrane, and the raw mean midline gray value for each image was normalized to the raw mean floor plate basement membrane value. The means of HH18, HH19, HH20, and HH21 normalized midline gray values were compared via Welch's unequal variances t-test. Finally, midline length was measured at HH19 as the distance between the tip of the endoderm and the notochord from the hindgut (where the 'bird's nest' of laminin deposition becomes longer than it is wide) to the cranial midgut (where the branches of the dorsal aorta fuse).

AMD3100-BODIPY diffusion was quantified by drawing an ROI across the DM bound by the dorsal aorta on top and the endoderm on the bottom. A profile plot of average gray values was obtained for each ROI for laminin and AMD3100-BODIPY separately. At HH19, the x-positions of laminin maxima corresponding to the left and right coelomic epithelium basement membrane and the midline basement membrane were used to divide the BODIPY intensity data into left and right DM compartments. At HH21 when no midline is present, left and right DM compartments were designated by dividing the distance between coelomic epithelial basement membranes in half. The separated left and right BODIPY data were then averaged to obtain a mean left DM and mean right DM BODIPY intensity value, which were compared to obtain the ratio of right vs. left DM BODIPY signal. All intensity measurements were normalized for each image by subtracting the average gray value of the notochord in the 488 channel. These ratios were compared via Welch's unequal variances t-test.

GM130 Golgi staining was used to assess cell polarity as described previously (*Welsh et al., 2013*). Five cell populations were evaluated: cells contacting the midline on the left ('left margin'), cells contacting the midline on the right ('right margin'), right mesenchymal cells not contacting the midline or coelomic epithelium, left coelomic epithelial cells, and rare cells observed between the double membrane of the midline itself ('midline cells') (*Figure 4D*, *Figure 4—figure supplement 1*). For each cell, the clockwise angle relative to vertical (0°) of the line drawn between the center of the nucleus and the Golgi apparatus (*Figure 4D*, inset) was recorded and plotted on an angle histogram in 20° bins with five biological replicates each. The trend of cell polarity in each cell population was assessed

using Rayleigh's test of uniformity using the 'circular' R package for circular statistics (https://r-forge.r-project.org/projects/circular/).

## Veiled chameleon husbandry, collection, and fixation

Veiled chameleon husbandry was performed at the Stowers Institute for Medical Research in the Reptiles and Aquatics Facility in accordance with the Institutional Animal Care and Use Committee approved protocol 2020-115, and as described previously (*Diaz et al., 2015b*; *Diaz et al., 2015a*; *Diaz et al., 2017*). following the protocols which are publicly available here: dx.doi.org/10.17504/protocols.io.bzhsp36e. Veiled chameleon eggs were collected at oviposition in the Reptiles and Aquatics Facility at Stowers Institute for Medical Research. Eggs were incubated in deli cups with moist vermiculite at a constant temperature of 28°C for 65–90 days to achieve desired staging. The eggs were cleared of large particles and wiped with RNaseZap wipes (Invitrogen AM9786) to minimize RNAse contamination. Clean eggs were candled to determine the position of the embryo under the leathery shell. We used fine scissors to cut a segment of the shell around the embryo and separate the embryo (attached to the shell) from the rest of the egg. The embryos were further separated from the shell and dissected out of the membranes in room temperature Tyrode's solution, made in DEPC-treated water. Subsequently, the embryos were fixed overnight at 4°C in 4% PFA in DEPC 1× PBS, then dehydrated through an ascending methanol series into 100% methanol and stored at –20°C for future analysis. Stages of embryonic development were determined as previously described (*Diaz et al., 2019*).

## Acknowledgements

We express deep gratitude to Drs. David Sherwood, Lydia Sorokin, Drew Noden, Gary Schoenwolf, and Peter Yurchenco for immensely helpful suggestions and wisdom. We deeply thank Dr. Ouathek Ouerfelli at Sloan Kettering for his services in synthesizing AMD3100-BODIPY. Sincere thanks go to Drs. John Grazul, Mariena Silvestry Ramos, and Shannon Caldwell (Cornell University), as well as Dr. Lihi Shaulov of the Technion Biomedical Electron Microscopy Center for instruction and technical assistance with Scanning Electron Microscopy. We appreciate the Cornell Imaging Core (Drs. Rebecca Williams and Johanna Dela Cruz) for training and maintenance of the microscopes. Thanks to Rachel Slater-Buchanan, Brittany Laslow, and Erica Butler for technical support. We thank Dr. Raphael Reuten for providing mouse Netrin 4 plasmid. Gratitude goes to Dr. Shing Hu for helpful feedback on the manuscript. We acknowledge Dr. Aravind Sivakumar for *Figure 6A* *Cxcl12* image and Dr. Bhargav Sanketi for *Figure 6B* *BMP4* image. This work was supported by the following grants: National Institute of Diabetes and Digestive and Kidney Diseases (grants R01 DK092776 and R01 DK107634 to NAK); the March of Dimes (grant 1-FY11-520 to NAK); NSF GRFP (DGE-1650441 to CD); NIH S10RR025502 for data collected on the Zeiss LSM 710 Confocal and NIH S10OD018516 for data collected on the inverted Zeiss LSM880 confocal/multiphoton microscope (i880); the Israel Science Foundation (grants 1463/16 and 1528/22 to TMS); the Israel Cancer Research Fund and the Rappaport Family Foundation to TMS; the Stowers Institute for Medical Research (PAT); and a K99 (HD114881) from the National Institute of Child Health and Human Development (NAS).

## Additional information

### Funding

| Funder | Grant reference number | Author |
| --- | --- | --- |
| National Institute of Diabetes and Digestive and Kidney Diseases | R01DK092776 | Natasza A Kurpios |
| National Institute of Diabetes and Digestive and Kidney Diseases | R01DK107634 | Natasza A Kurpios |
| March of Dimes Foundation | 1-FY11-520 | Natasza A Kurpios |

| Funder | Grant reference number | Author |
|---|---|---|
| Israel Science Foundation | 1463/16 | Thomas M Schultheiss |
| Israel Science Foundation | 1528/22 | Thomas M Schultheiss |
| Israel Cancer Research Fund | | Thomas M Schultheiss |
| Rappaport Family Foundation | | Thomas M Schultheiss |
| National Science Foundation | DGE-1650441 | Cora Demler |
| National Institute of Child Health and Human Development | HD114881 | Natalia Shylo |
| Stowers Institute for Medical Research | | Paul A Trainor |

The funders had no role in study design, data collection and interpretation, or the decision to submit the work for publication.

## Author contributions

Cora Demler, Conceptualization, Data curation, Formal analysis, Funding acquisition, Investigation, Validation, Visualization, Writing – original draft, Writing – review and editing, Methodology; John C Lawlor, Methodology, Supervision, Conceptualization, Formal analysis, Funding acquisition, Validation, Investigation, Writing – original draft; Ronit Yelin, Data curation, Methodology, Resources, Validation, Visualization, Supervision; Dhana Llivichuzcha-Loja, David Kim, Megan Stewart, Supervision, Formal analysis, Funding acquisition, Validation, Investigation; Lihi Shaulov, Writing – review and editing, Investigation; Frank K Lee, Resources, Investigation; Natalia Shylo, Writing – review and editing, Resources, Investigation; Paul A Trainor, Writing – review and editing, Resources; Thomas M Schultheiss, Methodology, Writing – review and editing, Conceptualization, Resources, Data curation, Investigation, Project administration, Writing – original draft; Natasza A Kurpios, Methodology, Writing – review and editing, Conceptualization, Resources, Data curation, Investigation, Visualization, Project administration, Writing – original draft

## Author ORCIDs

Cora Demler ⬥ https://orcid.org/0000-0003-3021-626X
John C Lawlor ⬥ https://orcid.org/0009-0003-4141-420X
Paul A Trainor ⬥ https://orcid.org/0000-0003-2774-3624
Natasza A Kurpios ⬥ https://orcid.org/0000-0002-1251-7395

Joint Public Review: https://doi.org/10.7554/eLife.89494.3.sa1
Author response https://doi.org/10.7554/eLife.89494.3.sa2

# Additional files

## Supplementary files

MDAR checklist

## Data availability

All data generated or analysed during this study are included in the manuscript and supporting files; source data files have been provided for *Figures 3, 4 and 6*.

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
