## [Editor Report · eLife Assessment]

This study reports the **fundamental** discovery of a novel structure in the developing gut that acts as a midline barrier between left and right asymmetries. Some of the evidence supporting the dynamics, composition, and function of this novel basement membrane in the chick is **solid**, some is even **convincing**, but investigation of its origin and impact on asymmetric organogenesis remains challenging and is not yet conclusive. This careful work is of broad relevance to patterning mechanisms, the importance of the extracellular matrix, and laterality disorders.

---

## [Referee Report · Joint Public Review]

When the left-right asymmetry of an animal body is established, a barrier that prevents the mixing of signals or cells across the midline is essential. Such midline barrier preventing the spreading of asymmetric Nodal signaling during early left-right patterning has been identified. However, midline barriers during later asymmetric organogenesis have remained largely unknown, except in the brain. In this study, the authors discovered an unexpected structure in the midline of the developing midgut in the chick. Using immunofluorescence, they convincingly show the chemical composition of this midline structure as a double basement membrane and its transient existence during the left-right patterning of the dorsal mesentery, that authors showed previously to be essential for forming the gut loop and guiding local vasculogenesis. Labelling experiments demonstrate a physical and chemical barrier function, to cell mixing and signal diffusion in the dorsal mesentery. Cell labelling and graft experiments rule out a cellular composition of the midline from dorsal mesenchyme or endoderm origin and rule out an inducing role by the notochord. Based on laminin expression pattern and Ntn4 resistance, the authors propose a model, whereby the midline basement membrane is progressively deposited by the descending endoderm. Observations of a transient midline basement membrane in the veiled chameleon suggest a conserved mechanism in birds and reptiles.

Laterality defects encompass severe malformations of visceral organs, with a heterogenous spectrum that remains poorly understood, by lack of knowledge of the different players of left-right asymmetry. This fundamental work significantly advances our understanding of left-right asymmetric organogenesis, by identifying an organ-specific and stage-specific midline barrier. The complexities of basement membrane assembly, maintenance and function are of importance in several other contexts, as for example in the kidney and brain. Thus, this original work is of broad interest.

Overall, reviewers refer to a strong and elegant paper discovering a novel midline structure, combining classic but challenging techniques, and well thought tools, to show the dynamics, chemical and physical properties of the midline. Reviewers also indicate that further work will be necessary to conclude on the origin and impact of the midline for asymmetric organogenesis. They acknowledge that this is currently technically challenging and that authors have made several attempts to answer these questions by different means. The article includes an interesting discussion about these points and the mechanism of midline breakdown.

---

## [Author Response]

The following is the authors’ response to the original reviews.

**Reviewer #1:**
Summary:Left-right asymmetry in the developing embryo is important for establishing correct lateralisation of the internal organs, including the gut. It has been shown previously that the dorsal mesentery (DM), which supports looping of the endodermal gut tube during development, is asymmetric with sharp delineation of left and right domains prior to gut looping. The authors set out to investigate the nature of the midline barrier that separates the left and right sides of the DM. They identify a transient basement membrane-like structure which is organised into two layers between the notochord and descending endoderm. In the time window when this basement membrane structure exists, there is no diffusion or cell mixing between the left and right sides of the DM, but once this structure starts breaking down, mixing and diffusion occur. This suggests it acts as a barrier, both physical and chemical, between left and right at the onset of gut lateralisation.Strengths:The authors identify a new midline structure that likely acts as a barrier to facilitate left and right separation during early organogenesis. This is an interesting addition to the field of laterality, with relevance to laterality-related disorders including heterotaxia, and may represent a gut-specific mechanism for establishing and maintaining early left-right asymmetry. The structure of this midline barrier appears to be an atypical basement membrane, comprising two adjacent basement membranes. The complexities of basement membrane assembly, maintenance, and function are of importance in almost all organismal contexts. Double basement membranes have been previously reported (for example in the kidney glomeruli as the authors note), and increasing evidence suggests that atypical basement membrane organisation or consideration is likely to be more prevalent than previously appreciated. Thus this work is both novel and broadly interesting.The data presented are well executed, using a variety of well-established methods. The characterisation of the midline barrier at the stages examined is extensive, and the data around the correlation between the presence of the midline barrier and molecular diffusion or cell mixing across the midline are convincing.Weaknesses:The study is rather descriptive, and the authors' hypotheses around the origins of the midline barrier are speculative and not experimentally demonstrated. While several potential origins of the midline are excluded raising interesting questions about the timing and cell-type-specific origin of the midline basement membrane, these remain unanswered which limits the scope of the paper.

We extend our appreciation to Reviewer #1 for their thoughtful and comprehensive evaluation of our work, recognizing the considerable time and effort they dedicated to our work. We agree that functional data would significantly strengthen our understanding of the midline barrier and its exact role during LR asymmetric gut development. However, we would like to note that repeated and diligent attempts to perturb this barrier were made using various strategies, such as in vivo laser ablation, diphtheria toxin, molecular disruption (Netrin 4), and enzymatic digestion (MMP2 and MMP9 electroporation) but we observed no significant effect or stable disruption of the midline. We acknowledge and accept this limitation and hope that our discovery will invite future investigations and perturbation of this novel midline structure.

For example, it is unclear whether the two basement membranes originally appear to be part of a single circular/spherical structure (which looks possible from the images) that simply becomes elongated, or whether it is indeed initially two separate basement membranes that extend.

We favor the hypothesis that the elongation of the preexisting small circular structure to an extended double membrane of relatively increased length would be unlikely without continued contribution of new basement membrane components. However, our attempts to label and trace the basement membrane of the endoderm using tagged laminins (LAMB1-GFP, LAMB1-His, and LAMC1-His), and more recently tagged nidogen constructs (NID1-GFP and NID1-mNG) have met with export issues (despite extensive collaboration with experts, Drs. Dave Sherwood and Peter Yurchenco). As such, it remains difficult to differentiate between the two possibilities suggested. We also believe this is an important question and will continue to investigate methods to trace it.

There is a substantial gap between the BMs at earlier stages before the endoderm has descended - is this a lumen, or is it filled with interstitial matrix?

Our preliminary studies indicate that the gap enclosed by the basement membranes in the early midline structure does have extracellular matrix present, such as fibrillin-2 (see Author response image 1). Also, the electron microscopy shown in Fig. 2 C’’ supports that the space between the notochord and endoderm has fibrillar matrix.

**Author response image 1. sa2fig1:** 

The authors show where this basement membrane does not originate from, but only speculate on its origin. Part of this reasoning is due to the lack of Lama1-expressing cells either in the early midline barrier before it extends, or in the DM cells adjacent to it. However, the Laminin observed in the midline could be comprised of a different alpha subtype for example, that wasn't assessed (it has been suggested that the Laminin antibody used in this study is not specific to the alpha-1 subunit, see e.g. Lunde et al, Brain Struct Funct, 2015).

We appreciate this comment and have tried other laminin RNA probes that showed similar lack of midline expression (Lama1, lama3, lama5). Importantly, the laminin alpha 1 subunit is a component of the laminin 111 heterotrimer, which along with laminin 511 is the first laminin to be expressed and assemble in embryonic basement membranes, as reviewed in Yurchenco 2011. Laminin 111 is particularly associated with embryonic development while laminins 511/521 become the most widespread in the adult (reviewed in Aumailley 2013). It is likely that the midline contains laminin 111 based on our antibody staining and the accepted importance and prevalence of laminin 111 in embryonic development. However, it is indeed worth noting that most laminin heterotrimers contain beta 1, gamma 1, or both subunits, and due to this immunological relation laminin antibody cross reactivity is certainly known (Aumailley 2013). As such, while laminin 511 remains a possibility as a component of the midline BM, our lama5 in situs have shown no differential expression at the midline of the dorsal mesentery (see Author response image 2), and as such we are confident that our finding of no local laminin transcription is accurate. Additionally, we will note that the study referenced by the Reviewer observed cross reactivity between the alpha 1 and alpha 2 subunits. Laminin 211/221 is an unlikely candidate based on the embryonic context, and because they are primarily associated with muscle basement membranes (Aumailley 2013). In further support, we recently conducted a preliminary transcriptional profile analysis of midline cells isolated through laser capture microdissection (LCM), which revealed no differential expression of any laminin subunit at the midline. Please note that these data will be included as part of a follow-up story and falls beyond the scope of our initial characterization.

Similarly, the authors show that the midline barrier breaks down, and speculate that this is due to the activity of e.g. matrix metalloproteinases, but don't assess MMP expression in that region.

This is an important point, as the breakdown of the midline is unusually rapid. Our MMP2 RNA in situ hybridization at HH21, and ADAMTS1 (and TS9) at HH19-21 indicates no differential activity at the midline (see Author response images 3 and 4). Our future focus will be on identifying a potential protease that exhibits differential activity at the midline of the DM.

**Author response image 3. sa2fig3:** 

**Author response image 4. sa2fig4:** 

The authors suggest the (plausible) hypothesis that the descent of the endoderm pulls or stretches the midline barrier out from its position adjacent to the notochord. This is an interesting possibility, but there is no experimental evidence to directly support this. Similarly, while the data supporting the barrier function of this midline is good, there is no analysis of the impact of midline/basement membrane disruption demonstrating that it is required for asymmetric gut morphogenesis. A more functional approach to investigating the origins and role of this novel midline barrier would strengthen the study.

Yes, we fully agree that incorporating functional data would immensely advance our understanding of the midline barrier and its crucial role in left-right gut asymmetry. However, our numerous efforts to perturb this barrier have encountered technical obstacles. For instance, while perturbing the left and right compartments of the DM is a routine and well-established procedure in our laboratory, accessing the midline directly through similar approaches has been far more challenging. We have made several attempts to address this hurdle using various strategies, such as in vivo laser ablation, diphtheria toxin, molecular disruption (Netrin 4), and enzymatic digestion (MMP2 and MMP9 electroporation). Despite employing diverse approaches, we have yet to achieve effective and interpretable perturbation of this resilient structure. We acknowledge this limitation and remain committed to developing methods to disrupt the midline in our current investigations. We again thank Reviewer #1 for the detailed feedback on our manuscript, guidance, and the time taken to provide these comments.

**Recommendations For The Authors:**
Using Laminin subunit-specific antibodies, or exploring the mRNA expression of more laminin subunits may support the argument that the midline does not derive from the notochord, endoderm, or DM.

As mentioned above, RNA in situ hybridization for candidate genes and a preliminary RNA-seq analysis of cells isolated from the dorsal mesentery midline revealed no differential expression of any laminin subunits.

Similarly, expression analysis of Laminin-degrading MMPs, and/or application of an MMP inhibitor and assessment of midline integrity could strengthen the authors' hypothesis that the BM is actively and specifically broken down.

Our MMP2 RNA in situ hybridization at HH21, and ADAMTS1 at HH19-21shows no differential expression pattern at the midline of the DM (see Author response image 3). We have not included these data in the revision, but future work on this topic will aim at identifying a protease that is differentially active at the midline of the DM.

Functionally testing the role of barrier formation in regulating left-right asymmetry or the role of endoderm descent in elongating the midline barrier would be beneficial. Regarding the former, the authors show that Netrin4 overexpression is insufficient to disrupt the midline, but perhaps overexpression of e.g. MMP9 prior to descent of the endoderm would facilitate early degradation of the midline, and the impact of this on gut rotation could be assessed.

Unfortunately, MMP9 electroporation has produced little appreciable effect. We acknowledge that the lack of direct evidence for the midline’s role in regulating left-right asymmetry is a shortcoming, but current work on this subject aims to define the midline’s function to LR asymmetric morphogenesis.

**Reviewer #2:**
When the left-right asymmetry of an animal body is established, the barrier that prevents the mixing of signals or cells across the midline is essential. The midline barrier that prevents the mixing of asymmetric signals during the patterning step has been identified. However, a midline barrier that separates both sides during asymmetric organogenesis is unknown. In this study, the authors discovered the cellular structure that seems to correspond to the midline in the developing midgut. This midline structure is transient, present at the stage when the barrier would be required, and composed of Laminin-positive membrane. Stage-dependent diffusion of dextran across the midline (Figure 6) coincides with the presence or absence of the structure (Figures 2, 3). These lines of indirect evidence suggest that this structure most likely functions as the midline barrier in the developing gut.

We extend our gratitude to Reviewer #2 for their thoughtful assessment of our research and for taking the time to provide these constructive comments. We are excited to report that we have now included additional new data on midline diffusion using BODIPY and quantification method to further support our findings on the midline's barrier function. While our data on dextran and now BODIPY both indirectly suggests barrier function, we aspire to perturb the midline directly to assess its role in the dorsal mesentery more conclusively. However, our numerous efforts to perturb this barrier have encountered technical obstacles. For instance, while perturbing the left and right compartments of the DM is a routine and well-established procedure in our laboratory, accessing the midline directly through similar approaches has been far more challenging. We have made several attempts to address this hurdle using various strategies, such as in vivo laser ablation, diphtheria toxin, molecular disruption (Netrin 4), and enzymatic digestion (MMP2 and MMP9 electroporation). Despite employing diverse approaches, we have yet to achieve effective and interpretable perturbation of this resilient structure. Moving forward, our focus is on identifying an effective means of perturbation that can offer direct evidence of barrier function.

**Recommendations For The Authors:**
(1) It would be much nicer if the requirement of this structure for asymmetric morphogenesis was directly tested. However, experimental manipulations such as ectopic expression of Netrin4 or transplantation of the notochord were not able to influence the formation of this structure (these results, however, suggested the mechanism of the midline formation in the gut dorsal mesentery). Therefore, it seems not feasible to directly test the function of the structure, and this should be the next issue.

We fully agree that the midline will need to be perturbed to fully elucidate its role in asymmetric gut morphogenesis. As noted, multiple attempts were ineffective at perturbing this structure. Extensive current work on this topic is dedicated to finding an effective perturbation method.

(2) Whereas Laminin protein was present in the double basement membrane at the midline, Laminin mRNA was not expressed in the corresponding region (Fig. 4A-C). It is necessary to discuss (with experimental evidence if available) the origin of Laminin protein.

As we have noted, the source of laminin and basement membrane components for the midline remains unclear - no local transcription and the lack of sufficiency of the notochord to produce a midline indicates that the endoderm to be a likely source of laminin, as we have proposed in our zippering endoderm model. We will note that Fig. 4A-C indicate that laminin is in fact actively transcribed in the endoderm. Currently, attempts to trace the endodermal basement membrane using tagged laminins (LAMB1-GFP, LAMB1-His, and LAMC1-His), and more recently tagged nidogen constructs (NID1-GFP and NID1-mNG) have met with export issues (despite extensive collaboration with experts, Drs. Dave Sherwood and Peter Yurchenco). Confirmation of our proposed endodermal origin model is a goal of our ongoing work.

(3) Figure 4 (cell polarity from GM130 staining): addition of representative GM130 staining images for each Rose graph (Figure 4E) would help. They can be shown in Supplementary Figures. Also, a graph for the right coelomic epithelium in Fig. 4E would be informative.

We have added the requested GM130 images in our Supplemental Figures (please refer to Fig. S4ABB’) and modified the main Fig. 4E to include a rose graph for the polarity of the right coelomic epithelium.

(4) Histological image of HH19 DM shown in Fig. 2J looks somehow different from that shown in Fig. 3F. Does Fig. 2J represent a slightly earlier stage than Fig. 3F?

Figure 2J and Figure 3F depict a similar stage, although the slight variation in the length of the dorsal mesentery is attributed to the pseudo time phenomenon illustrated in Figure 3J-J’’’. This implies that the sections in Figure 2J and Figure 3F might originate from slightly different positions along the anteroposterior axis. Nonetheless, these distinctions are minimal, and based on the dorsal mesentery's length in Figure 2J, the midline is likely extremely robust regardless of this minor pseudo time difference.

**Reviewer #3:**
Summary:The authors report the presence of a previously unidentified atypical double basement membrane (BM) at the midline of the dorsal mesentery (DM) during the establishment of left-right (LR) asymmetry. The authors suggest that this BM functions as a physical barrier between the left and the right sides of the DM preventing cell mixing and ligand diffusion, thereby establishing LR asymmetry.Strengths:The observation of the various components in the BM at the DM midline is clear and convincing. The pieces of evidence ruling out the roles of DM and the notochord in the origin of this BM are also convincing. The representation of the figures and the writing is clear.Weaknesses:The paper's main and most important weakness is that it lacks direct evidence for the midline BM's barrier and DM LR asymmetry functions.

We thank Reviewer #3 for their thoughtful and comprehensive evaluation of our work, recognizing the considerable time and effort they dedicated to assessing our study. We fully agree that incorporating functional data would immensely advance our understanding of the midline barrier and its crucial role in left-right gut asymmetry. However, several distinct attempts at perturbing this barrier have encountered technical obstacles. While our laboratory routinely perturbs the left and right compartments of the DM via DNA electroporation and other techniques, directly perturbing the midline using these methods is far more challenging. We have made diligent attempts to address this using various strategies, such as in vivo laser ablation, diphtheria toxin, molecular disruption (Netrin 4), and enzymatic digestion (MMP2 and MMP9 electroporation). However, we have not yet been able to identify a means of producing consistent and interpretable perturbation of the midline. We acknowledge this limitation and remain committed to developing methods to disrupt the midline in our current investigations.

**Recommendations For The Authors:**
Major:(1) We suggest the authors test their hypotheses i.e., physical barrier and proper LR asymmetry establishment by the midline BM, by disrupting it using techniques such as physical ablation, over-expression of MMPs, or treatment with commercially available enzymes that digest the BM.

As above, efforts involving physical ablation and MMP overexpression have not yielded significant effects on the midline thus far. Moving forward, investigating the midline's role in asymmetric morphogenesis will necessitate finding a method to perturb it effectively. In pursuit of progress on this critical question, we recently conducted laser capture microdissection (LCM) and RNA-sequencing of the midline to unravel the mechanisms underlying its formation and potential disruption. This work shows promise but it is still in its early stages; validating it will require significant time and effort, and it falls outside the scope of the current manuscript.

(2) Lefty1's role in the midline BM was ruled out by correlating lack of expression of the gene at the midline during HH19 when BM proteins expression was observed. Lefty1 may still indirectly or directly trigger the expression of these BM proteins at earlier stages. The only way to test this is by inhibiting lefty1 expression and examining the effect on BM protein localization.

We have added a section to discuss the potential of Lefty1 inhibition as a future direction. However, similar to perturbing global Nodal expression, interpreting the results of Lefty1 inhibition could be challenging. This is because it may not specifically target the midline but could affect vertebrate laterality as a whole. Despite this complexity, we acknowledge the value of such an experiment and consider it worth pursuing in the future.

(3) Using a small dextran-based assay, the authors conclude that diffusible ligands such as cxcl2 and bmp4 do not diffuse across the midline (Figure 6). However, dextran injection in this system seems to label the cells, not the extracellular space. The authors measure diffusion, or the lack thereof, by counting the proportion of dextran-labeled cells rather than dextran intensity itself. Therefore, This result shows a lack of cell mixing across the midline (already shown in Figure 2) rather than a lack of diffusion.

We should emphasize that the dextran-injected embryos shown in Fig. 6 D-F were isolated two hours post-injection, a timeframe insufficient for cell migration to occur across the DM (Mahadevan et al., 2014). We also collected additional post-midline stage embryos ten minutes after dextran injections - too short a timeframe for significant cellular migration (Mahadevan et al., 2014). Importantly, the fluorescent signal in those embryos was comparable to that observed in the embryos in Fig. 6. Thus, we believe the movement of fluorescent signal across the DM when the barrier starts to fragment (HH20-HH23) is unlikely to represent cell migration. More than a decade of DNA electroporation experiments of the left vs. right DM by our laboratory and others have never indicated substantial cell migration across the midline (Davis et al., 2008; Kurpios et al., 2008; Welsh et al., 2013; Mahadevan et al., 2014; Arraf et al. 2016; Sivakumar et al., 2018; Arraf et al. 2020; and Sanketi et al., 2022). This is also shown in our current GFP/RFP double electroporation data in Fig. 2 G-H, and DiI/DiO labeling data in Fig. 2 E-G. Collectively, our experiments suggest that the dextran signal we observed at HH20 and HH23 is likely not driven by cell mixing.

To further strengthen this argument, we now have additional new data on midline diffusion using BODIPY diffusion and quantification method to support our findings on the midline's function against diffusion (please refer to New Fig. 6H-M). Briefly, we utilized a BODIPY-tagged version of AMD3100 (Poty et al., 2015) delivered via soaked resin beads surgically inserted into the left coelomic cavity (precursor to the DM). The ratio of average AMD3100-BODIPY intensity in the right DM versus the left DM was below 0.5 when the midline is intact (HH19), indicating little diffusion across the DM (Fig. 6J). At HH21 when no midline remains, this ratio significantly rises to near one, indicating diffusion of the drug is not impeded when the midline basement membrane structure is absent. Collectively, these data suggest that the basement membrane structure at the midline forms a transient functional barrier against diffusion.

(4) Moreover, in a previous study (Mahadevan et al., Dev Cell., 2014), cxcl2 and bmp4 expression was observed on both the left and right side before gut closure (HH17, when midline BM is observed). Then their expression patterns were restricted on the left or right side of DM at around HH19-20 (when midline BM is dissociated). The authors must explain how the midline BM can act as a barrier against diffusible signals at HH-17 to 19, where diffusible signals (cxcl12 and bmp4) were localized on both sides.

We appreciate the Reviewer's invitation to clarify this crucial point. Early in dorsal mesentery (DM) formation, genes like Cxcl12 (Mahadevan et al., Dev Cell 2014) and Bmp4 (Sanketi et al., Science 2021) exhibit symmetry before Pitx2 expression initiates on the left (around ~HH18, Sanketi et al., 2021). Pitx2 then inhibits BMP4 (transcription) and maintains Cxcl12 (mRNA) expression on the left side. The loss of Cxcl12 mRNA on the right is due to the extracellular matrix (ECM), particularly hyaluronan (Sivakumar et al., Dev Cell 2018). Our hypothesis is that during these critical stages of initial DM asymmetry establishment, the midline serves as a physical barrier against protein diffusion to protect this asymmetry during a critical period of symmetry breaking. Although some genes, such as Pitx2 and Cxcl12 continue to display asymmetric transcription after midline dissolution (Cxcl12 becomes very dynamic later on – see Mahadevan), it's crucial to note that the midline's primary role is preventing protein diffusion across it, akin to an insurance policy. Thus, the absence of the midline barrier at HH21 does not result in the loss of asymmetric mRNA expression. We think its primary function is to block diffusible factors from crossing the midline at a critical period of symmetry breaking. We acknowledge that confirming this hypothesis will necessitate experimental disruption of the midline and observing the consequent effects on asymmetry in the DM. This remains central to our ongoing research on this subject.

(5) On page 11, lines 15-17, the authors mention that "We know that experimentally mixing left and right signals is detrimental to gut tilting and vascular patterning-for example, ectopic expression of pro-angiogenic Cxcl12 on the right-side results in an aberrant vessel forming on the right (Mahadevan et al., Dev Cell., 2014)". In this previous report from the author's laboratory, the authors suggested that ectopic expression of cxcl12 on the right side induced aberrant formation of the vessel on the right side, which was formed from stage HH17, and the authors also suggested that the vessel originated from left-sided endothelial cells. If the midline BM acts as a barrier against the diffusible signal, how the left-sided endothelial cells can contribute to vessel formation at HH17 (before midline BM dissociation)?

To address this point, we suggest directing the Reviewer to previously published supplemental movies of time-lapse imaging, which clearly illustrate the migration path of endothelial cells from left to right DM (Mahadevan et al., Dev Cell 2014). While the Reviewer correctly notes that ectopic induction of Cxcl12 on the right induces left-to-right migration, it's crucial to highlight that these cells never cross the midline. Instead, they migrate immediately adjacent to the tip of the endoderm (please also refer to published Movies S2 and S3). We observe this migration pattern even in wild-type scenarios during the loss of the endogenous right-sided endothelial cords, where some endothelial cells from the right begin slipping over to the left around HH19-20 (over the endoderm), as the midline is beginning to fragment, but never traverse the midline. We attribute this migration pattern to a dorsal-to-ventral gradient of left-sided Cxcl12 expression, as disrupting this pattern perturbs the migration trajectory (Mahadevan).

1. It is unclear how continuous is the midline BM across the anterior-posterior axis across the relevant stages. Relatedly, it is unclear how LR segregated the cells are, across the anterior-posterior axis across the relevant stages.

We refer the reviewer to Fig. 3J-K, in which the linear elongation of the midline basement membrane structure is shown and measured at HH19 in three embryos from the posterior of the embryo to the anterior point at which the midline is fragmented and ceases to be continuous. Similarly, Fig. S2 shoes the same phenomenon in serial sections along the length of the anterior-posterior (AP) axis at HH17, also showing the continuity of the midline. All our past work at all observed sections of the AP axis has shown that cells do not move across the midline as indicated by electroporation of DNA encoding fluorescent reporters (Davis et al. 2008, Kurpios et al. 2008, Welsh et al. 2013, Mahadevan et al. 2014, Sivakumar et al. 2018, Sanketi et al. 2022), and is shown again in Fig. 2 E-H. As noted previously, very few endothelial cells cross the midline at a point just above the endoderm (image above) when the right endothelial cord remodels (*Mahadevan et al. 2014*), but this is a limited phenomenon to endothelial cells and cells of the left and right DM are fully segregated as previously established.

Minor comments:(1) The authors found that left and right-side cells were not mixed with each other even after the dissociation of the DM midline at HH21 (Fig2 H). And the authors also previously mentioned that N-cadherin contributes to cell sorting for left-right DM segregation (Kurpios et al., Proc Natl Acad Sci USA., 2008). It could be a part of the discussion about the difference in tissue segregation systems before or after the dissociation of DM midline.

We appreciate this thoughtful suggestion. N-cadherin mediated cell sorting is key to the LR asymmetry of the DM and gut tilting, and we believe it underlies the observed lack of cell mixing from left and right DM compartments after the midline fragments. We have added a brief section to the discussion concerning the asymmetries in N-cadherin expression that develop after the midline fragments.

(2) Please add the time point on the images (Fig3 C, D, Fig 6A and B)

We have updated these figures to provide the requested stage information.

(3) The authors suggested that the endoderm might be responsible for making the DM BM midline because the endoderm links to DM midlines and have the same resistance to NTN4. The authors mentioned that the midline and endoderm might have basement membranes of the same "flavor." However, perlecan expression was strongly expressed in the midline BM compared with the endodermal BM. It could be a part of the discussion about the difference in the properties of the BM between the endoderm and DM midline.

Perlecan does indeed localize strongly to the endoderm as well as the midline. The HH18 image included in prior Fig. S3 B’, B’’ appears to show atypically low antibody staining in the endoderm for all membrane components. Perlecan is an important component for general basement membrane assembly, and the bulk of our HH18 and HH19 images indicate strong staining for perlecan in both midline and endoderm. Perlecan staining at the very earliest stages of midline formation also indicate perlecan in the endoderm as well, supporting the endoderm as a potential source for the midline basement membrane. We have updated Fig. S3 to include these images in our revision.

(4) The authors investigated whether the midline BM originates from the notochord or endoderm, but did not examine a role for endothelial cells and pericytes surrounding the dorsal aorta (DA). In Fig S1, Fig S2, and FigS3, the authors showed that DA is very close to the DM midline basement membrane, so it is worth checking their roles.

We fully agree that the dorsal aorta and the endothelial cords that originate from the dorsal aorta may interact with the midline in important ways. However, accessing the dorsal aorta for electroporation or other perturbation is extremely difficult. Additionally, the basement membrane of vascular endothelial cells has a distinct composition from a non-vascular basement membrane. Vascular endothelial cells produce only alpha 4 and alpha 5 laminin subunits but contain no alpha 1 subunit in any known species (*reviewed in DiRusso et al., 2017*). Thus, endothelial cell-derived basement membranes would not contain the alpha 1 laminin subunit that we used in our studies as a robust marker of the midline basement membrane. Additionally, no fibronectin is found in the midline basement membrane, while it is enriched in the dorsal aorta (see Supplemental Figure 3CC’C’’). We will briefly note that our preliminary data in quail tissue indicates that QH1+ cord cells (i.e. endothelial cells) sometimes exhibit striking contact with the midline along the dorso-ventral length of the DM, suggesting not an origin but an important interaction.

**Reviewer #4 (Recommendations For The Authors):**
Major comments:(1) The descending endoderm zippering model for the formation of the midline lacks evidence.

We have attempted to address this issue by introducing several tagged laminin constructs (LAMB1-GFP, LAMB1-His, LAMC1-His), and more recently tagged nidogen plasmids (NID1-GFP and NID1-mNG) to the endoderm via DNA electroporation to try to label the source of the basement membrane. Production of the tagged components occurred but no export was observed in any case (despite extensive collaboration with experts in this area, Drs. Dave Sherwood and Peter Yurchenco). This experiment was further complicated by the necessary large size of these constructs at 10-11kb due to the size of laminin subunit genes, resulting in low electroporation efficiency. We also believe this is an important question and are continuing to investigate methods to trace it.

The midline may be Ntn4 resistant until it is injected in the source cells.

Ntn4 has been shown to disrupt both assembling and existing basement membranes (*Reuten et al. 2016*). Thus, we feel that the midline and endodermal basement membranes’ resistance to degradation is not determined by stage of assembly or location of secretion.

Have you considered an alternative origin from the bilateral dorsal aorta or the paraxial mesoderm, which would explain the double layer as a meeting of two lateral tissues? The left and right paraxial mesoderm seem to abut in Fig. S1B-C and S2E, and is laminin-positive in Fig 4A'. What are the cells present at the midline (Fig.4D-E)? Are they negative for the coelomic tracing, paraxial or aortic markers?

We fully agree that alternate origins of the midline basement membrane cannot be ruled out from our existing data. We agree and have considered the dorsal aorta and even the endothelial cords that originate from the dorsal aorta. However, accessing the dorsal aorta for electroporation or other perturbation is extremely difficult. Importantly, the basement membrane of vascular endothelial cells has a distinct composition from a non-vascular basement membrane. Vascular endothelial cells produce only alpha 4 and alpha 5 laminin subunits but contain no alpha 1 subunit in any known species (*reviewed in Hallmann et al. 2005*). Thus, endothelial cell-derived basement membranes would not contain the alpha 1 laminin subunit that we used in our studies as a robust marker of the midline basement membrane. Note in Fig. 3 E-H that our laminin alpha 1 antibody staining does not label the aortae. Additionally, no fibronectin is found in the midline basement membrane, while it is enriched in the dorsal aorta (see Supplemental Figure 3CC’C’’). We will briefly note that our preliminary data in quail tissue indicates that QH1+ cord cells (i.e. endothelial cells) sometimes exhibit striking contact with the midline along the dorso-ventral length of the DM, suggesting not an origin but an important interaction. Moreover, at the earliest stages of midline basement membrane emergence, the dorsal aortae are distant from the nascent basement membrane, as are the somites, which have not yet undergone any epithelial to mesenchymal transition. Fig. S2G provides an example of an extremely early midline basement membrane without dorsal aorta or somite contact. S2G is from a section of the embryo that is fairly posterior in the embryo, it is thus less developed in pseudo-time and gives a window on midline formation in very early embryos.

(2) The importance of the midline is inferred from previously published data and stage correlations but will require more direct evidence. Can the midline be manipulated with Hh signaling or MMPs?

We agree that direct evidence in the form of midline perturbation will be critically required. As previously noted, our numerous efforts to perturb this barrier have encountered technical obstacles. For instance, while perturbing the left and right compartments of the DM is a routine and well-established procedure in our laboratory, accessing the midline directly through similar approaches has been far more challenging. We have made several attempts to address this hurdle using various strategies, such as in vivo laser ablation, diphtheria toxin, molecular disruption (Netrin 4), and enzymatic digestion (MMP2 and MMP9 electroporation). Despite employing diverse approaches, we have yet to achieve effective and interpretable perturbation of this resilient structure. Targeting Hh signaling between the endoderm and notochord is a good idea and we will continue these efforts. Thanks very much.

Minor comments:- Please add the species in the title.

We have altered the title as follows: “An atypical basement membrane forms a midline barrier during left-right asymmetric gut development in the chicken embryo.”

- The number of observations in Fig2, Fig3A-B, 4A-C, G-H, S1, S3 is lacking.

We have added the requested n numbers of biological replicates to the legends of the specified figures.

- Please annotate Fig 3J to show what is measured in K.

We have modified Fig. 3J to include a dashed bar indicating the length measurements in Fig. 3K.

- Please provide illustrations of Fig 4E.

We have added a representative image of GM130 staining to the supplement.

- If laminin gamma is the target of Ntn4, its staining would help interpret the results of Ntn4 manipulation. Is laminin gamma present in different proportions in the different types of basement membranes, underlying variations in sensitivity?

Laminin is exported as a heterotrimer consisting of an alpha, beta, and gamma subunit. Laminin gamma is therefore present in equal proportions to other laminins in all basement membranes with a laminin network. Several gamma isoforms do exist, but only laminin gamma 1 will bind to laminin alpha 1, which we use throughout this paper to mark the midline as well as nearby basement membranes that are sensitive to Ntn4 disruption. Thus, gamma laminin proportions or isoforms are unlikely to underlie the resistance of the midline and endodermal basement membranes to Ntn4 (*reviewed in Yurchenco 2011*).

- Please comment: what is the red outline abutting the electroporated DM on the left of Fig5B?

The noted structure is the basement membrane of the nephric duct – we added this information to Fig. 5B image and legend.

- The stage in Fig 6A-B is lacking.

We have added the requested stage information to Fig. 6.

- Please comment on whether there is or is not some cell mixing Fig 2H, at HH21 after the midline disappearance. Is it consistent with Fig. 6E-F which labels cells?

More than a decade of DNA electroporation experiments of the left vs. right DM by our laboratory and others have never indicated dorsal mesentery cell migration across the midline (Davis et al., 2008; Kurpios et al., 2008; Welsh et al., 2013; Mahadevan et al., 2014; Arraf et al. 2016; Sivakumar et al., 2018; Arraf et al. 2020; and Sanketi et al., 2022). This is also shown in our current GFP/RFP double electroporation data in Fig. 2 G-H, and DiI/DiO labeling data in Fig. 2 E-G. Cell mixing does not occur even after midline disappearance, most likely due to asymmetric N-cadherin expression on the left side of the DM (Kurpios et al., 2008). The sparse, green-labeled cells observed on the right side in Fig. 2H are likely a result of DNA electroporation - the accuracy of this process relies on the precise injection of the left (or right) coelomic cavity (precursor to the gut mesenchyme including the DM) and subsequent correct placement of the platinum electrodes.

Based on these data, we strongly feel that cellular migration is not responsible for the pattern of dextran observed in Fig. 6E-F, especially in light of the N-cadherin mediated segregation of left and right. We will also note that there is no significant difference between dextran diffusion at HH19 and HH20, only a trend towards significance. Additionally, we would like to note that the dextran-injected embryos were isolated two hours post-injection, which we do not believe is sufficient time for any cell migration to occur across the DM. We also collected additional post-midline stage embryos ten minutes after dextran injections (data not shown), too short a timeframe for significant cellular migration, and the fluorescent signal in those embryos was comparable to that represented in the embryos in Fig. 6. Thus, we believe the movement of fluorescent signal across the DM observed when the barrier starts to fragment at HH20 and HH23 is unlikely to represent movement of cells.

To further strengthen this argument, we now have additional new data on midline diffusion using BODIPY and quantification method to support our findings on the midline's function against diffusion (please refer to New Fig. 6H-M). Briefly, we utilized a BODIPY-tagged version of AMD3100 (Poty et al., 2015) delivered via soaked resin beads surgically inserted into the left coelomic cavity (precursor to the DM). The ratio of average AMD3100-BODIPY intensity in the right DM versus the left DM was below 0.5 when the midline is intact (HH19), indicating little diffusion across the DM (Fig. 6J). At HH21 when no midline remains, this ratio significantly rises to near one, indicating diffusion of the drug is not impeded when the midline basement membrane structure is absent. Collectively, these data suggest that the basement membrane structure at the midline forms a transient functional barrier against diffusion.

- 'independent of Lefty1': rephrase or show the midline phenotype after lefty1 inactivation.

We agree with this comment and have rephrased this section to indicate the midline is present “at a stage when Lefty1 is no longer expressed at the midline.”

We again would like to extend our sincere gratitude to our reviewers and the editors at eLife for their dedicated time and thorough evaluation of our paper. Their meticulous attention to detail and valuable insights have strengthened our data and provided further support for our findings.